# Interannual Variability of Terpenoid Emissions in an Alpine City

**Lisa Kaser[1], Arianna Peron[1], Martin Graus[1], Marcus Striednig[1], Georg Wohlfahrt[2], Stanislav Juráň[3], Thomas Karl[1]**

[1]Department of Atmospheric and Cryospheric Sciences, University of Innsbruck, Innrain 52f, 6020 Innsbruck, Austria

[2]Department of Ecology, University of Innsbruck, Sternwartestrasse. 15, 6020 Innsbruck, Austria

[3] Global Change Research Institute of the Czech Academy of Sciences, Bělidla 986/4a, 603 00 Brno, Czech Republic

*Correspondence to*: Thomas Karl (thomas.karl@uibk.ac.at) and Lisa Kaser (kaser.lisa@gmail.com)

**Abstract.** Terpenoid emissions above urban areas are a complex mix of biogenic and anthropogenic emission sources. In line with previous studies we found that summertime terpenoid fluxes in an alpine city were dominated by biogenic sources. Inter-seasonal emission measurements revealed consistency for monoterpenes and sesquiterpenes, but a large difference in isoprene between the summers 2015 and 2018. Standardized emission potentials for monoterpenes and sesquiterpenes were 0.12 nmol m$^{-2}$ s$^{-1}$ and 3.0 x 10$^{-3}$ nmol m$^{-2}$ s$^{-1}$ in 2015 and 0.11 nmol m$^{-2}$ s$^{-1}$ and 3.4 x 10$^{-3}$ nmol m$^{-2}$ s$^{-1}$ in 2018, respectively. Observed isoprene fluxes were almost three times higher in 2018 than in 2015. This factor decreased to 2.3 after standardizing isoprene fluxes to 30°C air temperature and photosynthetic active radiation (PAR) of 1000 μmol m$^{-2}$ s$^{-1}$. Based on emission model parameterizations, increased leaf temperatures can explain some of these differences, but standardized isoprene emission potentials remained higher in 2018, when a heat wave persisted. These data suggest a higher variability of interannual isoprene fluxes than for other terpenes. Potential reasons for the observed differences such as emission parameterization, footprint changes, water stress conditions and tree trimming are investigated.

## 1 Introduction

Biogenic and anthropogenic volatile organic compounds (BVOCs, AVOCs) in the atmosphere can contribute to surface air pollution both due to their influence on tropospheric ozone formation and due to their potential to act as precursors for secondary organic aerosol (Derwent et al., 1996, Fehsenfeld et al., 1992, Fuentes et al., 2000, Goldstein et al., 2009, Laothawornkitkul et al., 2009, Riipinen et al., 2012). BVOCs are playing a particularly important role globally, as their emission strength is estimated to be 10 times larger than AVOCs (Guenther et al., 2012, Piccot et al., 1992). Also, many BVOCs are characterized as highly reactive (Atkinson and Arey, 2003, Fuentes et al., 2000), resulting in rapid peroxy radical chemistry important for ozone and ultra-fine particle formation processes (Simon et al., 2020). Of the total global BVOC emissions, terpenes dominate, with 50% attributed to isoprene, 15% to monoterpenes and about 0.5% to sesquiterpenes (Guenther et al., 2012). In predominantly isoprene-emitting forests isoprene was found responsible for 50-100% of the tropospheric ozone production (Duene et al., 2002, Tsigaridis and Kanakidou, 2002, Poisson et al., 2001). In coniferous forests monoterpene and sesquiterpene emissions often dominate (Johansson and Janson, 1993, Thunis and Cuvelier, 2000, Juráň et al., 2017). It has been shown that RO$_2$ self-reactions of monoterpenes and sesquiterpenes can rapidly create highly oxidized matter (HOM) and are a key player for new particle formation (NPF) events in forests under low NOx conditions (Simon et al., 2020). In urban environments where the mixture of BVOCs and AVOCs is more complex, several recent studies point out the importance of biogenic emissions for local air quality (Simon et al., 2019, Bonn et

al., 2018, Churkina et al., 2017, Ren et al., 2017, Papiez et al., 2009, Chameides et al., 1988) and that
the BVOC influence is especially high during summertime heat waves (Churkina et al. 2017).
Particularly in summer, biogenic sources are dominating in urban environments. E.g., Yadav et al.
(2019) found an increased importance of biogenic isoprene in an urban site in western India during pre-
monsoon season when temperatures and PAR were high, Hellen et al. (2012) found a strong biogenic
influence on isoprene and monoterpene concentrations in Helsinki in July. Summertime isoprene in two
large Greek cities was determined by PMF to mainly (60-70%) originate from vegetation (Kaltsonoudis
et al. 2016). Yang et al. (2005) showed a strong seasonal and daily cycle in isoprene, attributing it
therefore to biogenic sources in an urban region in Taiwan. Borbon et al (2002) showed that biogenic
sources strongly superimpose the traffic emissions of isoprene in summer in an urban area in France.
Wagner and Kuttler (2014) found that during summer afternoons in an urban area in Germany
anthropogenic influences on isoprene concentrations were negligible. Chang et al. (2014) and Wang et
al. (2013) showed that in a tropical-subtropical metropolis biogenic contributions overwhelmed
anthropogenic contributions of isoprene in summer and that biogenic sources started to dominate in all
seasons above a threshold temperature of 17-21°C. Whereas all the studies cited above were based on
concentration measurements where the influence can be both local and regional and strongly modulated
by atmospheric dilution, the following studies were based on eddy covariance flux tower sites. At
temperatures over 25°C more than 50% of the isoprene flux was found to be biogenic in origin in
London with a mean daytime flux of 0.18 mg $m^{-2}$ $h^{-1}$ (Langford et al. 2010). Similarly, Valach et al.
(2015) in a different study in London found a mean daytime flux of 0.2 mg $m^{-2}$ $h^{-1}$. Kota et al. (2014)
found a daytime median flux of 2.1 mg $m^{-2}$ $h^{-1}$ over Houston, Texas and contributed it to mostly
biogenic sources. Park et al (2010) found also in Houston, Texas a daytime isoprene flux of 0.7 mg $m^{-2}$
$h^{-1}$. Rantala et al. (2016) found that 80% of the measured 10 ng $m^{-2}$ $s^{-1}$ summer daytime isoprene flux
near Helsinki could be contributed to biogenic sources by comparing emissions at low and high
temperatures.
While there is evidence for urban trees to have positive influence on urban environments such as
mitigating the urban heat island effect, sequestering $CO_2$ and particles as well as via storm water
interception (Escobedo et al., 2011, Connop et al., 2016, Livesley et al., 2016), BVOC emissions of
urban trees and their subsequent effect on air pollution are very plant-species dependent (Corchnoy et
al., 1992, Steinbrecher et al., 2009, Fitzky et al., 2019) and should be taken into account when planting
urban trees (Calfapietra et al., 2013, Churkina et al., 2015, Ren et al., 2017). Emerging evidence that
isoprene derived $RO_2$ competes with $RO_2$ radicals from higher molecular weight terpenes in the
formation of new particles highlights the need to study emissions in different environments (Berndt et
al. 2018).
Few studies characterize the interannual changes of BVOCs and even fewer such studies are available
in urban environments. Vaughan et al. (2017) report airborne flux measurements over South Sussex of
two consecutive summers showing different isoprene fluxes that can be explained by different
temperature and cloud cover conditions. Warneke et al. (2010) tried to explain the measured interannual
differences of a factor of 2 in fluxes of isoprene and monoterpene over Texas by temperature, drought
effects or influences from changes in leaf area index (LAI). Palmer et al. (2006) found a maximum of
20-30% interannual difference in isoprene emissions using satellite-based isoprene quantification from
formaldehyde measurements over North America. A model study by Steinbrecher et al. (2009) found
only a 10% annual difference in biogenic emissions from cold to hot years. Gulden et al. (2007) found
that, on a regional scale, variations in leaf biomass density driven by variations in precipitation are
together with temperature and shortwave radiation variations the most important factors for variations in
BVOC emissions. Tawfik et al. (2012) found in a model study that interannual variation of isoprene
emission is strongest in July with temperature and soil moisture explaining 80% of the variations,
whereas the influences of variations in photosynthetic active radiation (PAR) and LAI were negligible.
In a three-year study over a northern hardwood forest, Pressley et al. (2005) found that total cumulative
isoprene fluxes varied only by 10%.
Given the current lack of multi-year urban VOC flux measurements and our limited understanding of
the interannual variability of biogenic and anthropogenic emission sources, the objective of the present
study was to quantify the interannual variation of the urban ecosystem-atmosphere exchange of the
three major isoprenoids, isoprene, monoterpenes, and sesquiterpenes, and to analyze the underlying
drivers. We hypothesized (i) that the exchange of these BVOCs can be attributed largely to the spatio-
temporal variability of biogenic sources and (ii) that differences in environmental forcings are the main
drivers of interannual variability. To address these hypotheses, urban eddy covariance BVOC flux
measurements during two growing seasons above the city of Innsbruck (Austria) are blended with
bottom-up emission estimates based on a process-based model and a detailed urban tree inventory.

## 2 Materials and methods

### 2.1 Field site and instruments

VOC concentrations and flux measurements were conducted during two comparable summer periods
(July10-September 9 2015 & July 27-September 2 2018) close to the city center of Innsbruck on the
rooftop of one of the tallest buildings in the area. The data record in 2018 is continuous, in 2015 the
data record has a gap between July 31 and August 03. Details on the Innsbruck Atmospheric
Observatory (IAO) measurement site and instrument performance were published by Karl et al. (2018)
and Striednig et al. (2020). Therefore, we give here only a short summary of the study location and
measurement details. The measurement location ($47^{\circ}15'51.66''$ N, $11^{\circ}23'06.82''$ E) is shown in Figure 1A
on a 2000x2000m map surrounding the site. The dominant wind direction at the IAO is from the NE
during the daytime and from the SW during nighttime (Karl et al. 2020, Striednig et al. 2020). Within
500 m from IAO, the mean building height is 17.3 m whereas the modal building height of about 19 m
corresponds to the 5–7 story buildings, which are more important in terms of their form drag. For this
reason, the displacement height, $z_d$, is estimated as 13.3 m (0.7 m $\times$ 19 m). The roughness length, $z_0$, is
1.6 m.
3D sonic wind, $CO_2$, and $H_2O$ were measured with a CPEC200 (Campbell Scientific) eddy covariance
system at a sampling frequency of 10 Hz on a tower on top of the building 42 m above street level. In
2015 the tower was at a provisional location at the north of the building; the heading direction of the
sonic anemometer was 76°. Flow distortions for westerly winds due to the building and the support
structure cannot be excluded. In the course of the establishment of the IAO lab the CPEC200 the inlets
were moved ~50 m to the southern edge of the building with an anemometer heading of 129° and
minimal flow disturbances. For comparability isoprenoid fluxes in this study are limited to the
northeastern sector of [0°,120°] in both years.
A heated inlet line led from the tower to a close-by laboratory hosting a PTR-QiTOF-MS instrument
(IONICON Analytik, Sulzer et al. 2014), which allows for the acquisition of full, high-resolution mass-
spectral information at 10 Hz. Residence time of air samples in the turbulently purged teflon inlet line
(Teflon PFA, ¼" ID x 12.7m heated at 30°C) is about 0.4 seconds to keep wall loss and chemical
transformation of isoprenoids negligible. Both summers the PTR-QiTOF-MS was operated in $H_3O^+$
mode with standard drift tube conditions of 112 Townsend (E/N electric field strength). Regular
instrument calibrations and zeroing revealed typical acetone and isoprene sensitivities of 1550 and 950
Hz/ppbv respectively.
Incident PAR was calculated from short wave radiation measured by a pyranometer (Schenk 8101,
Schenk, Wien) applying the relationship derived by Jacovides et al. (2003) (PAR/short wave radiation
~0.46 during summer daytime conditions).
Precipitation data were collected 400 m south of our field site by a  tipping bucket precipitation gauge
(MPS TRWS 503) and a precipitation monitor (Thies 5.4103.10.000), mounted at 1.5 m above a grass
surface, both operated by Zentralanstalt für Meteorologie und Geodynamik (ZAMG, Austrian Met-
Service) at the station Innsbruck Universität (WMO SYNOP number 11320).
Due to the lack of directly measured city-scale soil moisture data, plant available soil moisture for 2015-
2019 was retrieved as the SMAP level 4 3-hourly 9 km rootzone soil moisture product (Reichle et al.
2018) via the AppEEARS interface (https://lpdaacsvc.cr.usgs.gov/appeears/). Due to the large spatial
footprint of this product, the corresponding data will only be used to interpret interannual differences in
precipitation on soil moisture.
**2.2 Eddy covariance fluxes**
This study focuses on biogenic fluxes collected during summer 2015 and summer 2018. The presented
eddy covariance flux measurements are used to constrain BVOC flux parameterizations. Biogenic
emissions, in particular isoprene, are strongly light- and temperature-driven. As a consequence we
selected daytime flux data. During daytime the flux footprint density points towards the east sector
imposed by the local valley wind system. In order to test BVOC emission parameterizations we
therefore selected daytime hours ( 06:00-18:00 local time) and mean wind directions from 0°-120°.
Data with wind direction from the south and exceeding a wind speed > 10 m/s were excluded as they
can be attributed to foehn events, for which we believe current footprint density calculations bear too
much uncertainty in an urban setting. Eddy covariance fluxes were calculated using a MATLAB® code
described by Striednig et al. (2020). Figure S1 shows the co-spectral response of the PTR-QiTOF-MS
and inlet system. The loss of covariance of isoprenoids signals with vertical windspeed due to lowpass
filtering is less than 4% (see Spectral analysis in Supplemental Information)
As a QA/QC criteria for fluxes we implemented a combination of steady state filter of the respective
scalar, the integral turbulence characteristics test of the wind components and flow sector filtering,
similar to the combination described in Chapter 4.2.5. in Foken (2017) with a required overall quality
class of 6 or lower. According to Foken (2017), classes 1-6 can be used for long-term measurements of
fluxes without limitations. Implementing these QA/QC criteria reduced the available flux data by 29%
and 11% in 2015 and 2018, respectively.
The footprint density representing the relative contributions of an air mass sample arriving at the flux
tower was calculated following Kljun et al. (2015).
Constraints on the lifetime of reactive terpenes: Turbulent time scales (100 s) can be the order of
chemical time scales of some monoterpenes which can react fast with ozone. We calculate the chemical
loss by the following equation: $c(t)/c0=\exp(-t_{turb}/t_{chem})$ where $t_{turb}$ is the turbulent time scale and $t_{chem}$ the
chemical time scale. The turbulent time scale was obtained from the ratio of the measurement height
(H) over the friction velocity (H/u*). For typical turbulent time scales of 100 s, reaction with OH can be
neglected.
Further, our analysis of emissions is primarily focused on the interpretation of daytime fluxes, when
NO3 radical chemistry plays a minor role compared to ozone. Ozone follows the expected diurnal cycle
for an urban area (30-50 ppbv mixing ratios). Since we do not have speciated terpene fluxes, we
performed a sensitivity study (e.g. estimating realistic bounds) assuming a fraction of the total
sesquiterpene (or monoterpene) flux was composed by the most reactive compound (rSQT and rMT).
For sesquiterpenes, for example, we can take the estimated rate constant for ozone and beta
caryophyllene: 1.2e-14 cm3/molecules/s.  A typical compositional mix of sesquiterpenes was reported
by Sakulyanontvittaya et al., 2008), who assessed reactive terpene fractions between 36-50%. Typical
reaction rates of less reactive sesquiterpenes (nrSQT) (e.g. cedrene, longifolene: Atkinson et al., 1994)
are on the order of 1 to 10 x 1e-17 cm3/molecules/s. Taking these boundary conditions  gives a realistic
range of the reacted fraction of measured SQT fluxes. Similarly we can do the analysis for
monoterpenes, where the fraction of reactive terpenes (rMT) such as ocimene is typically lower (e.g. 10
- 15% - Sakulyanontvittaya et al., 2008,). For comparison, trans-beta-ocimene, one of the most reactive
monoterpenes known to be emitted from plants, has a reaction rate constant of 2.6e-14 cm3/molecules/s.
Figure S2 and S3 in the supplemental information show the non-reacted flux for total sesquiterpenes
due to reaction with ozone assuming a 36 to 64 and a 50 to 50 mix (rSQT to nrSQT). With these
scenarios daytime reductions of total sesquiterpenes fluxes due to chemistry would be on the order of
30-45%. For monoterpene fluxes we calculate losses on the order of 12% (Figure S4).

### 2.3 Emission standardization of fluxes

**Big leaf model for standardization of surface fluxes**: We standardized isoprene eddy covariance fluxes $E_{0,ISO}$, to a temperature of 303.15 K and PAR of 1000 µmol m$^{-2}$ s$^{-1}$ using a model described in detail by Guenther et al. (2006): $E_{ISO} = E_{0,ISO} * \gamma_T * \gamma_P$, where $\gamma_T$ and $\gamma_P$ are temperature- and light-dependent coefficients respectively containing current and past (24h and 240h) conditions. Monoterpene and sesquiterpene emissions are often dominated by temperature. Originally the temperature dependence has been described as: $E_{MT} = E_{0,MT} * C_{T,MT}$ and $E_{SQT} = E_{0,SQT} * C_{T,SQT}$ where $C_T$ is a temperature dependent factor (e.g. Guenther et al. 1994). Some monoterpene and sesquiterpene emissions have also been reported to be produced de novo and can therefore show a light dependent emission behavior (e.g. Staudt and Seufer, 1995). The light dependent portion is included in updated emission algorithms (e.g. Guenther et al., 2012, equation 3-6), where the light dependent portion is modeled in analogy to isoprene, and the light independent fraction is incorporated according to Guenther et al. 1994. The light dependent fraction for monoterpenes varies between 0.2 and 0.8, and for sesquiterpenes it is currently assumed to be 0.5. The temperature and light parameterization was calculated using equation 3 - 11 from Guenther et al., (2012) who prescribed a 50% light dependent fraction for SQT emissions. For Monoterpenes we take the average light dependent fraction from Guenther et al., 2012 (i.e. 50%), since we do not have speciated MT fluxes.

**MEGAN 5 layer model**: In order to investigate the sensitivity of isoprene emissions to the emission model framework we also setup a 5 layer canopy model according to Guenther et al. (2006). The setup was used to conduct a sensitivity experiment to study potential inter-seasonal changes in isoprene emissions between 2015 and 2018 based on different model formulations. For the sensitivity run the model was constrained by measured radiative fluxes, sensible and latent heat fluxes. We prescribed a LAI of 1 to account for sparse vegetation and mimic a sunleaf dominated scenario, with a mean sunlit fraction of 64% (40-95%).

Direct LAI measurements are not available for this study. Both campaigns were conducted in similar time frame within the year which should lead to comparable leaf age. No early senescence in either year was reported by the city gardeners.

### 2.4 Bottom-up emission potentials

**City tree inventory:** An inventory of all trees planted by the city municipality is available for the city of Innsbruck, Austria containing location, tree species, diameter at breast height and height. However, this inventory does not include trees from private gardens. Therefore, all accessible trees from private gardens were identified and added to the existing tree inventory in an area 1000x1000m surrounding the observatory. This will in the following be referred to as the study area.. The location of the trees from the city inventory (41 %) and private gardens (59 %) in the study area are shown in Figure 1A. Within the study area a total of 1904 registered trees distributed across 129 tree species were counted and it is estimated that these cover > 90 % of the available trees . A list of the 44 most abundant tree species, where the species count in the study area was 6 or more, is given in Table 1.

**Emission potentials:** Literature values of plant-species specific emission potentials of isoprene and monoterpene, in µg compound g$^{-1}$ dry-weight h$^{-1}$ standardized to 303.15 K and PAR 1000 µmol m$^{-2}$ s$^{-1}$ were assigned to the 44 most abundant species in the study area. This includes all tree species with an occurrence larger than 6 individuals within the 60% footprint density and accounts for ~90% of the total counted trees. Emission potential assignment was based, if available, on the detailed work by Stewart et al. (2003). Other emission potentials were taken from other literature and if more than one literature value was available, an average was taken. All species, emission potentials and references thereof are

shown in Table 1. Sesquiterpene (SQT) emission potentials were taken from Karl et al. (2009) and if not
reported therein the average value of 0.1 µg compound g$^{-1}$ dry-weight h$^{-1}$ was assigned.

**2.5 Relative ISO, MT, SQT emission ratio maps**

To generate emission ratio maps, the study area was divided into a 100 m by 100 m grid and tree
species were counted in each grid tile and multiplied by their emission potential listed in Table 1. The
resulting map in units of µg compound g$^{-1}$ dry-weight h$^{-1}$ neglects the actual, but unknown, amount of
dry leaf weight of each individual tree.
Due to the unknown amount of emitting leaf material, it is difficult to compare bottom-up estimates
from this method with direct eddy covariance flux measurements. A more robust comparison is possible
when relative emission maps are investigated such as ISO/MT, ISO/SQT and SQT/MT. For this we first
added up all individual tree emission factors in each tile (e.g. $ISO_{tile} = \sum ISO_{tree}$) and then divided
these by the tile emission factors e.g. ISO$_{tile}$/MT$_{tile}$. For simplicity this is in the following called
ISO/MT. This is a bottom-up ISO/MT ratio expected at the measurement site. The authors acknowledge
that  leaf age, phenology, and LAI or individual trees affect this ratio but both are unknown for the tree
inventory and are therefore a source of uncertainty of this estimate. Doubling and halving the emission
potential of the highest 20 emitters resulted in average study area emission ratios changes on the order
of 5-15% giving an estimate of the robustness of this analysis.

**3 Results and discussion**

**3.1 Flux footprint, light & temperature conditions**

The flux footprint density at the IAO is shown in Figure 1A and 1B for 2015 and 2018, respectively.
Flux footprint density lines from 30-90% are plotted on a map of 2000 m x 2000 m surrounding the flux
tower location. 60% of the flux footprint density lay, in both years, entirely within the study area (1000
m x 1000 m). The relative contribution of the land cover types within the study are was similar in both
years with 40-41% buildings, 23% paved areas, 25-28% roads, 5% trees, 5% short vegetation, and <1%
water. Within the 60% flux footprint density area in 2015 and 2018 lay 148 and 89 individual trees of
the tree inventory distributed over 33 and 24 tree species, respectively. Combining the tree inventory
with literature values on basal emission factors (Table 1) and the footprint density calculated for each
tree location revealed that 60% and 70% of the bottom-up isoprene emissions arriving at the flux tower
were from 12 trees in 2015 and 2018 respectively. These were trees closest to the footprint density
maximum and trees with high isoprene basal emission factors. The tree species were *Populus nigra*,
*Platanus acerifolia, Sophora japonica*, and *Quercus robur.* As the 60% footprint density area was
smaller in 2018 compared to 2015, the relative importance of the emission of these trees was higher in
2018 than in 2015. Bottom-up monoterpene emissions were distributed more evenly among different
tree species: 19 trees in the study area accounted for ~50% of the bottom-up MT emissions arriving at
the flux tower. The most important species were *Aesculus carnea*, *Pinus sylvestris*, *Larix decidua,* and
*Acer platanoides*. Sesquiterpene bottom-up emissions were even more equally distributed over the tree
species: 38 trees accounted for 50% and 60% of bottom-up SQT emissions arriving at the flux tower in
2015 and 2018 respectively. *Betula pendula* and *Sophora japonica* contributed 20% and 12% to the
emissions arriving at the tower in 2015 and 22% and 19% in 2018. Diurnal cycles of PAR and air
temperature, two of the strongest biogenic emission drivers, are shown in Figure 1C and 1D,
respectively. While PAR was very similar during the two summers, mean air temperatures in 2018 were
2K higher during daytime and 1.5K higher during nighttime compared to 2015. The higher temperatures
in 2018 coincided with an intense heat wave. Monthly average temperatures in August 2018 were 3K
above the climatological mean values (1981-2010).

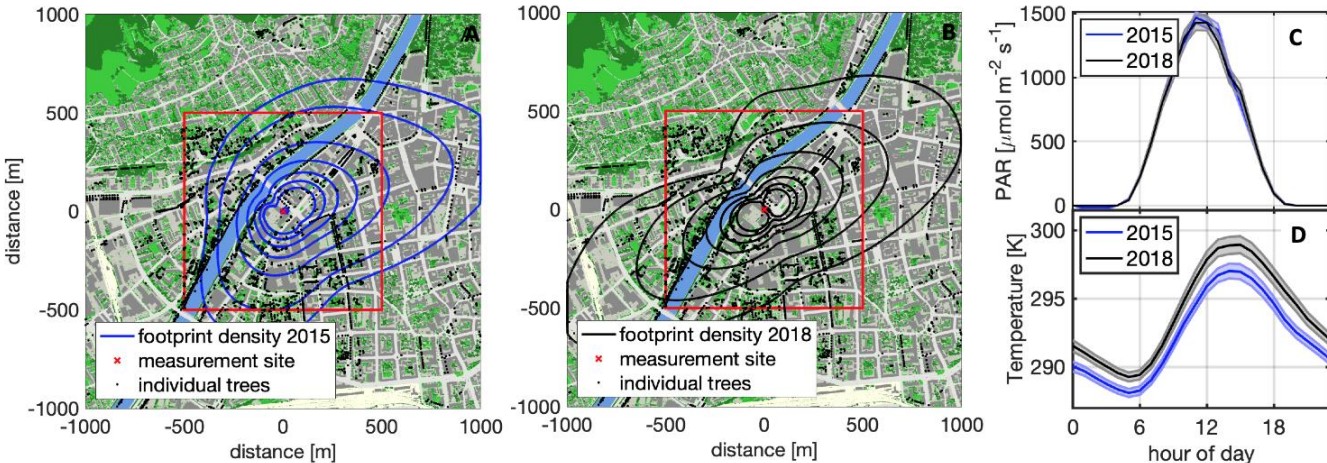

**Figure 1: A)** Map surrounding the Innsbruck Atmospheric Observatory (indicated with a red cross in the center) depicting trees, short vegetation, water, roads, paved areas and buildings in dark green, light green, blue, white, light grey and dark grey respectively. Black dots represent individual trees from the city tree inventory. The study area is indicated with the red rectangle. 2015 footprint density lines from 30%-90% are plotted as blue lines. **B)** Same as map as A with 2018 footprint density lines in black. **C)** Diurnal cycle of average and standard error of PAR in 2015 (blue) and 2018 (black). **D)** Diurnal cycle of average and standard error of ambient temperature 2015 (blue) and 2018 (black) Maps were created in Matlab (www.mathworks.com) and are based on OpenStreetMap (https://www.openstreetmap.org/copyright) under the CC BY 3.0 AT license.

### 3.2 Two summers of urban isoprenoid fluxes

Karl et al. (2018) showed that isoprene and monoterpene at this measurement site are linked to biogenic processes. Figure 2 A-C show the average diurnal cycles of isoprene, monoterpene and sesquiterpene fluxes. Mean daytime maxima of isoprene fluxes were 0.4 nmol m$^{-2}$ s$^{-1}$ and 1.2 nmol m$^{-2}$ s$^{-1}$ in 2015 and 2018 respectively. The large interannual difference and its potential reasons are discussed further in section 3.3. Anthropogenic contributions to isoprene emissions from traffic were on the one hand estimated using the COPERT emission model (https://www.emisia.com/utilities/copert/) and 1,3 butadiene as a proxy. We use the ratio of 1,3 butadiene to isoprene from road tunnel studies (Reimann et al., 2000) and multiply this to the modeled 1,3 butadiene to benzene ratio. There is no significant modeled difference between warm and cold seasons because unsaturated hydrocarbons as well as benzene primarily originate from combustion related emissions. Relative to benzene we calculate that anthropogenic isoprene emissions contribute on the order of 5% during daytime (Fig. S5 for the summer season. At night the contribution can be larger (e.g. up to 20%) as biogenic emissions decrease more rapidly than benzene fluxes. On the other hand we used the measured winter-time isoprene/benzene flux ratio, which revealed a conservative limit of 20% due to anthropogenic origin. Overall isoprene emissions are dominated by biogenic emissions at this site. This is in good accordance with previous studies conducted in urban environments (Kota et al. 2014, Park et al 2010, Rantala et al. 2016).

Maximum average daytime monoterpene fluxes were 0.13 nmol m$^{-2}$ s$^{-1}$ and 0.18 nmol m$^{-2}$ s$^{-1}$ for 2015 and 2018, respectively, and average daytime sesquiterpene fluxes were 5 x 10$^{-3}$ nmol m$^{-2}$ s$^{-1}$ in both years.

The theoretical temperature and light parameters are plotted vs. the observed fluxes in Figure 2 D-F based on the MEGAN big leaf approach (Guenther et al., 2006, Guenther et al., 2012.). The slope of the fit parameters represents the standardized (303.15 K and 1000 PAR) emission factors. The slopes in Fig. 2. D-F can be interpreted as standardized fluxes, removing the variability due to current and past temperature and light conditions and allows for interannual comparison as well as comparison to other studies. Standardized isoprene fluxes were 0.26±0.02 nmol m$^{-2}$ s$^{-1}$ and 0.67±0.02 nmol m$^{-2}$ s$^{-1}$ in 2015 and 2018 respectively. The interannual difference is further discussed in section 3.3. Isoprene fluxes from both years were lower than what Rantala et al. (2016) found for an urban flux site in Helsinki, where the standardized emission potential was 125 ng m$^{-2}$ s$^{-1}$ (eq. to: 1.8 nmol m$^{-2}$ s$^{-1}$). The Helsinki flux site had a larger vegetation cover of 38-59% compared to our study area, where the vegetation

cover was estimated to be 10% within the flux footprint. Park et al. (2010) reported a standard emission rate of isoprene of 0.53 mg m$^{-2}$ h$^{-1}$ (eq. to: 2.2 nmol m$^{-2}$ s$^{-1}$) over Houston, Texas, which is higher than both our 2018 and 2015 measurements. This is potentially due to a higher vegetation cover in Houston as well as strong isoprene-emitting oaks within the footprint of the measurement site. Valach et al. (2015) reported a daytime average flux in August of 0.3 mg m$^{-2}$ h$^{-1}$ (eq. to: 1.2 nmol m$^{-2}$ s$^{-1}$) at an urban site in London and Acton et al. (2020) a summer daytime average isoprene flux of 4.6 nmol m$^{-2}$ s$^{-1}$ at an urban site in Beijing, both however cannot be directly compared to our measurements as their values were not standardized to temperature and PAR.

Average daytime standardized monoterpene fluxes were, with 0.04 and 0.05 nmol m$^{-2}$ s$^{-1}$ in 2015 and 2018, respectively, relatively similar between the two summers. Average daytime standardized sesquiterpene fluxes were over a magnitude smaller than standardized monoterpene fluxes and were comparable between the two summers with mid-day values on the order of 3.0 x 10$^{-3}$ nmol m$^{-2}$ s$^{-1}$ and 3.5 x 10$^{-3}$ nmol m$^{-2}$ s$^{-1}$ in 2015 and 2018 respectively. Both monoterpene and sesquiterpene flux measurements could be underestimated due to loss with reaction to ozone. The values given here could be underestimated by 10% for monoterpenes and 35-45% for sesquiterpenes (see section 2.2). Monoterpene and sesquiterpene fluxes measured at lower temperatures (280K-295K) were higher than the predicted values based on biogenic emission parameterizations (data not shown). This could be an indication that at lower temperatures other, non-biogenic sources contributed to monoterpene and sesquiterpene fluxes at this site. At temperatures higher than 295K, MT and SQT fluxes followed known temperature dependencies. To test this hypothesis we considered footprint variations and relative distributions between grasses and trees, which were minor. Variations in flux footprint and a relative distribution with higher grassland MT emissions can be excluded as an explanation for MT and SQT excursions. Instead we find that the residual of non-explained MT and SQT fluxes correlates well with aromatic fluxes. We find a significant positive correlation (R2 ~0.75; RMSE: 0.006204) of the residual MT flux with the benzene flux (Fig S5). It suggests that emission of volatile chemical products (VCPs) (e.g. Gkatzelis et al. 2021) is the most likely explanation for MT and SQT flux enhancements that are not being reproduced by biogenic emission parameterizations.

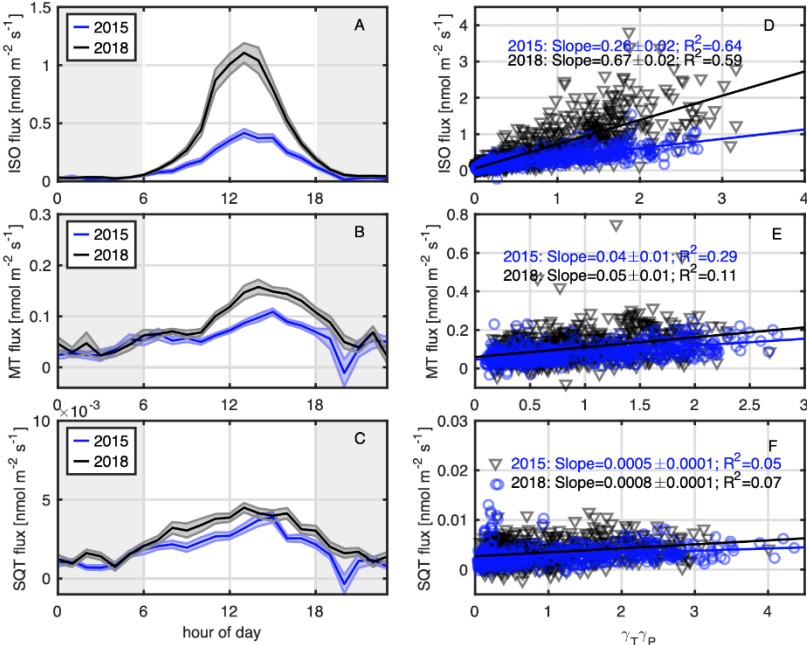

**Figure 2: Diurnal cycles of average isoprene (A), monoterpene (B) and sesquiterpene (C) fluxes for the summers 2015 (blue) & 2018 (black), shaded areas indicate the standard error. Night-time fluxes are shown here for completeness of the diurnal cycle but gray shaded areas indicate that this data was not used for further analysis. (D-F) Daytime (6:00-18:00) isoprene, monoterpene, and sesquiterpene fluxes are plotted vs. theoretical temperature and light dependencies (Guenther et al. 2006, Guenther et al. 2012) including T24, T240, P24, P240. 2015 data are depicted in blue and 2018 data in black. The lines indicate a linear fit with fit**

**parameters displayed within the plot. The slope of the fit parameter represents the standardized (303.15 K and 1000 PAR)**
**emission factors.**

## 3.3 Isoprene flux anomaly

The isoprene flux difference measured between the two summers of 2015 and 2018 is shown in Figure 2
A. Daytime maximum isoprene fluxes in 2018 were up to 2.7 times higher than in 2015. Isoprene fluxes
are temperature dependent (Guenther et al. 1993), light dependent (Monson and Fall, 1989) as well as
past 24h and 240h temperature and light conditions play a role (e.g. Guenther et al., 2006). These
theoretical temperature and light parameters are plotted vs. the observed isoprene flux in Figure 2D
based on the MEGAN big leaf approach (Guenther et al., 2006). Even after including both actual and
past temperature and light parameters the difference in isoprene fluxes between the two summers could
not be resolved and standardized emission factors were still a factor 2.3 higher in 2018 than in 2015.
Figure 3B shows that the difference was increasing with higher temperature and higher PAR values.

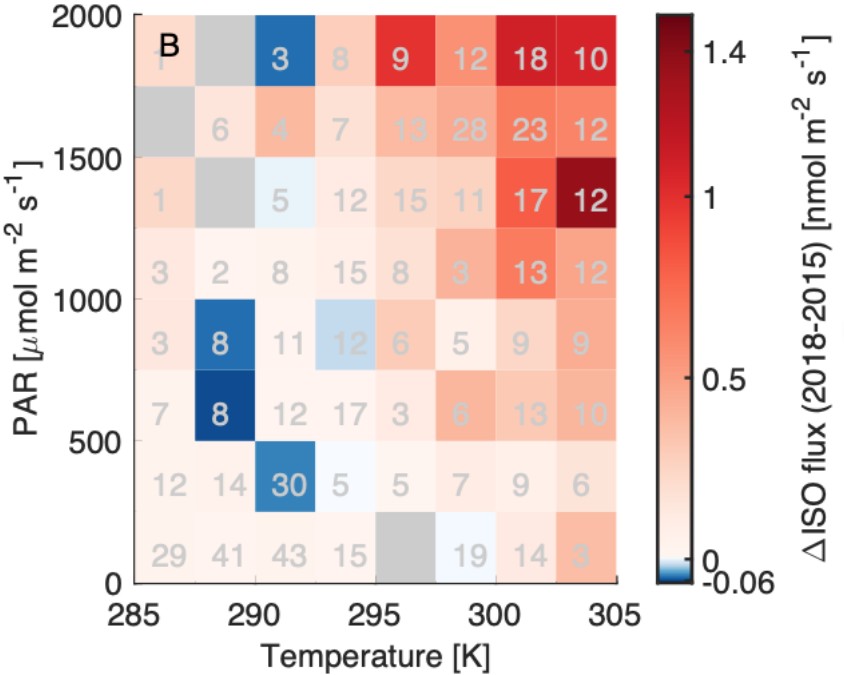

**Figure 3: Isoprene flux differences between 2018 and 2015 binned by temperature and PAR, positive differences are shown in red,**
**negative in blue and bins with no available data are colored grey. Grey numbers in the temperature/PAR fields indicate the**
**number of observations for each temperature/PAR value pair.**
In contrast to monoterpene and sesquiterpene fluxes, which exhibited comparable emission potentials
between the two years and are mainly driven by evaporative emissions from storage reservoirs (e.g.
Kesselmeier and Staudt, 1999), it remains a puzzle why the isoprene emission potential was
substantially higher in 2018 compared to 2015. As neither actual temperature and light dependencies
nor 24h and 240h past temperature and light could fully explain the observed differences in isoprene
fluxes, we investigated the following potential reasons: a) variation in the flux footprint, b) tree
trimming, c) water availability/drought, and d) emission parameterization.
a) Figure 1 (left and middle panel) shows differences in the flux footprint densities between 2015
and 2018. Possible reasons for this are a change in flux tower position between the two years by
~50 m, as well as different meteorological conditions. In 2015, for westerly winds the flow
regime may have been affected by the support structure and the building and consequently the
analysis of isoprenoid fluxes was limited to the northeastern wind sector of [0°,120°]. Median
daytime wind speed and direction in 2018 (affected by a heat wave) are similar to those in 2015

(Table S1). Median sonic temperature, sensible heat flux and friction velocity (see Table S1) were higher in 2018 resulting in stronger turbulent vertical transport with mostly friction velocity being responsible for the differences of the flux footprint density function between 2015 and 2018 (Fig 1). Multiplying the footprint density at each tree location with the basal emission factor of each tree species revealed a potential difference of 24% higher isoprene emissions in 2018 than in 2015. Even though the actual leaf area of each individual tree is not known and therefore neglected, this 24% of potential emission difference due to footprint density changes cannot explain the factor of 2.3 in observed fluxes between the two years. Also growth of juvenile trees between the study years is unlikely to play a significant role, as just 8 % of the strong isoprene emitters were younger than 5 years in 2015. This analysis assumes that the trees from the tree inventory were responsible for the majority of measured isoprene fluxes and that they were more important than emissions from short vegetation (e.g. lawn). Further supporting evidence that the flux footprint change cannot fully explain the observed differences derives from the fact that both monoterpenes and sesquiterpenes did not show significant inter-annual variations in their normalized emission potentials.

b) A second possible explanation for the isoprene flux difference could be differences in LAI in the two seasons, for example due to pruning, early leaf senescence or insect/pathogen damage. Personal communications from city gardeners revealed that of the trees most important for isoprene emissions in the study area (*Populus nigra*, *Populus alba*, *Quercus robur*) only poplar trees were cut differently in 2015 than in 2018. In 2015 only dead wood was removed from the poplars, whereas trees were cut more substantially in 2018. This would however lead to an expected smaller flux in 2018 than in 2015 due to reductions in leaf area. No observations on early leaf senescence or leaf damage by insects/pathogens were reported by the city gardeners during both study years.

c) A third possible explanation is that the growing season of 2018 was exceptionally dry with lower-than-average precipitation and large-sale, satellite-derived rootzone soil moisture (Figure 4A and 4B). Concurrent water flux observations, however, shown in Fig. 4C, indicate that on average the 2018 daytime summer water flux was 0.2 mmol m$^{-2}$ s$^{-1}$ higher than in 2015. Also the total surface water vapor conductance was 50 mmol m$^{-2}$ s$^{-1}$ higher in 2018 than in 2015. Higher water fluxes observed in 2018 agree with anecdotal reports of city trees being artificially watered throughout the summer. Water fluxes in urban areas (maximum Bowen ratios observed in Innsbruck: 6 (Karl et al., 2020)) are generally very low (e.g. 5-6 times lower) when compared to measurements over purely vegetated surfaces, and therefore notoriously difficult to interpret. As such we cannot exclude the possibility of processes other than evapotranspiration from city trees contributing to higher water fluxes observed in 2018. An obvious explanation is that a significant water runoff during extensive watering operations resulted in increased evaporation over hot asphalt and other non-vegetated surfaces, leading to higher water fluxes in 2018. Water was also applied to asphalt surfaces more frequently during mornings to minimize the effect of urban aerosol pollution. The cumulative precipitation for July, August and September 2015 was 340 mm, and 258 mm for 2018. When taking just overlapping campaign duration data (July 27 - Sept 2), the cumulative precipitation was 158 mm in 2015 and 155 mm in 2018. The precipitation data confirms an overall drier meteorological summer in 2018. It is well established that isoprene production in plants can decouple from photosynthesis during periods of drought and can be sustained by alternative metabolic carbon sources (e.g. Bertin & Staudt, 1996; Pegoraro et al., 2004a,b; Fortunati et al., 2008; Genard-Zielinski et al., 2014; Potosnak et al., 2014; Wu et al., 2015). The exact reason for biochemical regulation of isoprene emissions during drought is not fully unraveled, but has been suggested to represent a response for coping with heat stress (Loreto et al., 1998). Isoprene fluxes were observed to increase during the very early onset of drought conditions. For example, Seco et al. (2015) reported an increase in the ecosystem scale isoprene emission potential about one month before significant changes in pre-dawn leaf water potential were observed, but when $CO_2$ uptake was already decreasing. Additionally, they observed that the closing of stomata had a bigger effect on $CO_2$ than water

fluxes, because gradual increases of vapor pressure deficit during the evening offset reduced leaf conductance. Isoprene is not controlled by stomata and would not be influenced by any changes in stomatal opening. In addition, their canopy scale observations suggested a shift of the temperature maximum of isoprene emissions towards higher temperatures from pre-drought to drought conditions. Otu-Larbi et al. (2019) reported a 2.5 fold increase in the isoprene emission potential during the same 2018 heat wave in a UK oak forest. They observed a strong temperature dependence of isoprene concentrations during the heat wave and discuss potential causes such as leaf temperature or rewetting enhanced emissions. While we do not have representative soil moisture data available for this study, we looked at precipitation data. Otu-Larbi et al. (2020) observed large increases in within- and above-canopy isoprene mole fractions in response to rainfall events after a 6-week drought in a temperate broadleaf forest, which they interpreted to result from enhanced isoprene emissions following the rewetting. We consider rewetting events an unlikely explanation for the observed higher isoprene fluxes in 2018 because, even though rainfall was reduced by half compared to 2015, rain-free time intervals were quite short (between 2 and 7 days) and thus no pronounced rewetting occurred after a long dry period. In fact, the isoprene flux time series suggests lower emissions following rain events. We would like to note that both mono- and sesquiterpene emissions are also controlled by stomatal conductance which could be expected to affect emission rates during drought periods (see e.g. Niinemets and Reichstein, 2003a & b). We did not observe significant differences of mono- and sesquiterpene fluxes between the seasons.

d) We also examined the impact of emission model framework on isoprene emissions. Due to the lack of directly measured soil moisture data, which would be hard to interpret in an urban context, the drought effect was not included in the emission model parameterization. Precipitation (Fig. 4A) and large-scale satellite-derived soil moisture data (Fig. 4B) suggest 2018 being drier than 2015, corresponding to a significant heat wave in the summer of 2018. Severe drought conditions would reduce isoprene emissions further and therefore could not explain an increased isoprene emission potential in 2018. The fact that evaporative water fluxes however were comparable between 2015 and 2018 (and if at all were somewhat higher in 2018), suggest that the trees might not have undergone a severe drought episode in both years. Mild drought has been observed to lead to increases of isoprene emissions (e.g. Otu-Larbi et al., 2019). To investigate relative changes between emission model frameworks we also set up a MEGAN 5-layer canopy model (Guenther et al., 2006) for different scenarios. We recognize that the concept of an LAI for the 5 layer model is based on the assumption of a homogeneous vegetation distribution. The resulting fraction of sun vs shade leaves for urban vegetation might therefore not be fully constrained without complex 3D radiative transfer simulations in urban situations with sparsely distributed vegetation. The prescribed setup however was chosen to mimic a high sunlight fraction of the biomass with an overall fraction of 64%. The model setup was in turn only used to see whether differences between 2015 and 2018 could theoretically be explained by a high sunlight fraction or different temperature response curves. We observed that a shift in $T_{opt}$ towards higher temperatures helped minimize the observed difference between the two years (e.g. 10% to 40%) best. So, for example $T_{opt}$ set to 313 K could explain about half of the flux enhancement. This would leave predicted isoprene emission fluxes underestimated by about 50% in 2018. The combination of footprint (24%) and $T_{opt}$ (50%) could bring the isoprene emission potential between 2015 and 2018 to within 37% uncertainty.

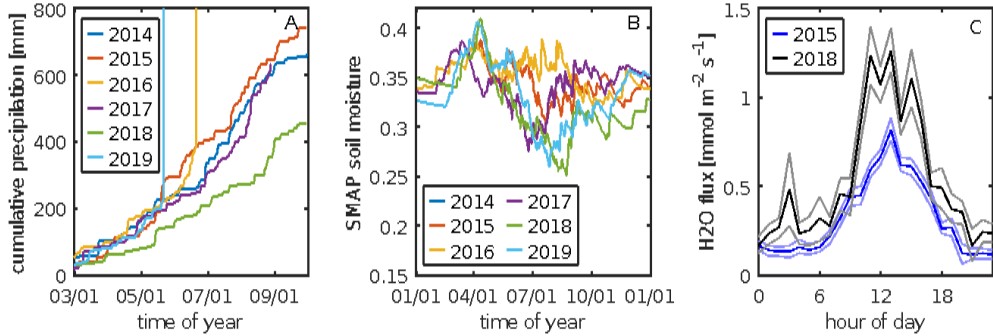

Figure 4: A) cumulative precipitation for the growing seasons of 2014-2019 B) annual SMAP satellite soil moisture of the rootzone from 2014-2019. C) diurnal cycle of water fluxes measured in 2015 (blue) and 2018 (black).

### 3.4 Top-down flux and bottom-up isoprenoid emission ratios

Standardized top-down flux ratios were calculated to allow for a better comparison with bottom-up emission estimates based on literature values of branch level emissions and a city tree inventory. Top-down (eddy covariance) $ISO_S/MT_S$ flux ratios were on the order of 5 in 2015 and 12 in 2018, again revealing a strong difference between the two years. Top-down $MT_S/SQT_S$ flux ratios were in the order of 30-40 before factoring in losses of sesquiterpenes due to reactions with ozone. Factoring in the upper bound of chemical loss correction, $MT_S/SQT_S$ flux ratios could have been as low as 12-16. Top-down $ISO_S/SQT_S$ flux ratios lay on the order of 190 in 2015 and 380 in 2018, which was mostly caused by the difference in $ISO_S$ flux between the two years. The lower bounds of the $ISO_S/SQT_S$ flux ratios due to fast reaction of sesquiterpene with ozone were 80 and 150 for 2015 and 2018, respectively. Branch-level standardized emissions are collected from the literature in Table 1 and are used to calculate a bottom-up emission map shown in Figures 5 A-C. The 2018 footprint area (Fig. 1A) and therefore footprint density was different to 2015. Multiplying bottom-up emission estimates with footprint density functions, the theoretically expected ISO/MT, MT/SQT and ISO/SQT ratios in 2015 were 3.6, 5.1 and 18.7 respectively. Multiplying the 2018 footprint density, the values were slightly different with 4.2, 4.6 and 19.2 for ISO/MT, MT/SQT and ISO/SQT ratios respectively. The bottom-up ISO/MT emission ratio was close to the top-down ratio of 2015. This indicates again that 2018 was an exceptional year. In contrast, the bottom-up MT/SQT and ISO/SQT emission ratios were significantly lower than both summers top-down measured flux ratios. Even after accounting for the chemical loss of sesquiterpene before it reached the point of measurement at the top of the building, the bottom-up estimates were still higher than the top-down measured flux ratios. Literature values for leaf level sesquiterpene emissions are rare and were for many species estimated in Table 1. Further extensive studies on sesquiterpene standardized emissions for a large variety of plant species is needed to close the gap between bottom-up emission ratios and top-down flux ratio estimates.

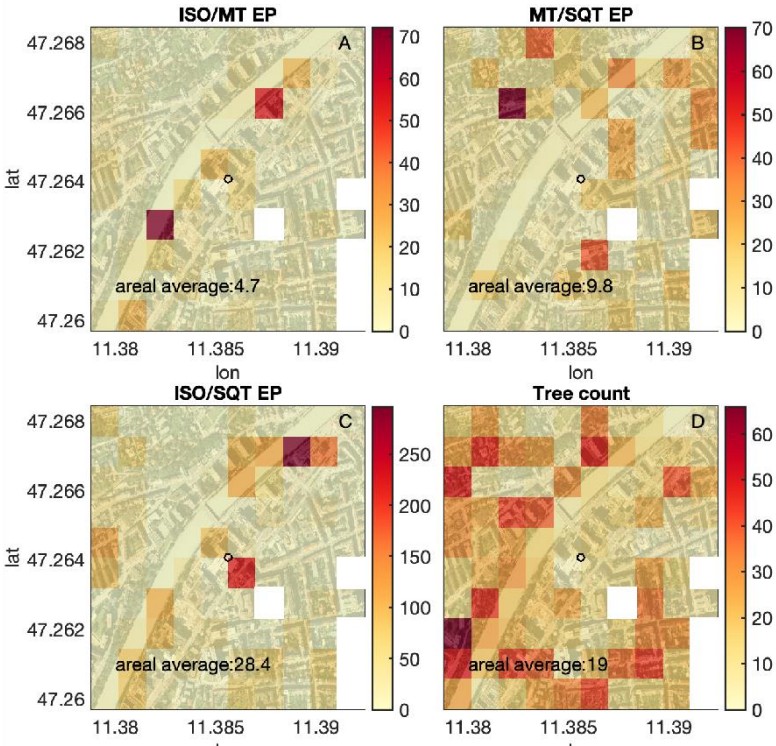

**Figure 5: A-C: Bottom-up estimates of standardized ISO/MT, MT/SQT and ISO/SQT emission ratios based on literature values (see Table 1). D: Tree count. Maps were created in Matlab (www.mathworks.com) and are based on OpenStreetMap (https://www.openstreetmap.org/copyright) under the CC BY 3.0 AT license.**

## 4 Summary

In this study we found a strong correlation of isoprene fluxes with temperature as well as isoprene fluxes following the previously observed leaf-level light dependency. Assuming the same correlation between isoprene and benzene fluxes in early spring before the start of the vegetation period and the summer months results in a maximum of 20-30% influence of anthropogenic sources on isoprene emissions during both 2015 and 2018 summer measurement periods. A PMF analysis at this site (Karl et al. 2018) has previously revealed two biogenic factors: one light- and temperature-dependent for isoprene and a second mostly temperature-dependent including monoterpenes and sesquiterpenes. Bottom-up emission estimates based on a city tree inventory and emission factors from literature showed a reasonable agreement to standardized ISO/MT flux ratios and an underestimation of standardized MT/SQT and ISO/SQT flux ratios. Interannual comparison of biogenic fluxes revealed up to three times higher isoprene fluxes in 2018, when a heat wave persisted, than in 2015. Monoterpene fluxes were an order of magnitude lower than isoprene fluxes and sesquiterpene fluxes were another order of magnitude lower than monoterpene fluxes, however both summer fluxes were comparable for these two terpenoid classes after standardization. Our findings show a higher interannual variability of isoprene emissions compared to monoterpenes and sesquiterpenes. Normalizing isoprene fluxes to standard light conditions did not fully remove the interannual difference, but decreased the factor from 3 to 2.3. The difference increased with higher temperature and higher PAR values. Analysis of footprint, precipitation and a coarse-scale satellite-based soil moisture product as a proxy for plant water availability, and pruning activity differences of the two summers did not completely resolve the observed differences in isoprene fluxes. Detailed analysis using standard emission modeling concepts suggested a higher-than-expected variation of urban isoprene emission potentials during the heat wave in 2018. While water flux measurements did not indicate a severe drought in 2018, the effect of an intense heatwave in 2018 (2K higher temperatures on average compared to 2015), likely resulted in enhanced isoprene emissions. Isoprene emissions during drought stress have been grouped into two

distinct phases (Niinemets, 2010, Potosnak et al., 2014), and can be enhanced under pre-drought conditions (Seco et al., 2015, Otu-Larbi et al., 2019). Enhanced leaf temperatures (e.g. Potosnak et al., 2014) can explain part of the variance in isoprene emissions, but significant differences remained. In addition to the leaf temperature effect, Tattini et al., (2015) reported an upregulation of isoprene emissions during drought stress as antioxidant defense in *Platanus x acerifolia* plants. Here a change of $T_{opt}$ towards a higher temperature optimum could explain another 50% of the observed isoprene emission flux difference between 2015 and 2018. In conjunction with changes in flux footprints (24%) these two effects could account for about ~75% of the difference. If generalized, our observations suggest distinct differences that urban trees experience, possibly due to significantly altered environmental conditions (e.g. stresses, light and temperature environment). Vegetation in urban areas is exposed to a variety of different atmospheric conditions, for example the urban heat island effect, high levels of $NO_y$, heavy metal deposition or high loadings of aerosols (e.g. black soot). Isoprene emissions have been linked to the plant's nitrogen metabolism (e.g. Rosenstiel et al., 2008), where higher leaf nitrate can lead to lower isoprene emissions. Nitrogen dioxide concentrations have been falling in Innsbruck and were 20% lower in 2018 than in 2015. Effects of air pollutants on leaf surface characteristics and senescence were also reported in the past (Jochner et al. 2015; Honour et al., 2009), but a quantitative understanding of the impacts on isoprene emissions remains unclear. Our observations suggest that more work is needed to improve our understanding of urban biogenic isoprene emissions.

## Acknowledgements

This work was primarily funded by the Hochschulraum-Strukturmittel (HRSM) funds sponsored by the Austrian Federal Ministry of education, science and research (https://www.bmbwf.gv.at/), the EC Seventh Framework Program (Marie Curie Reintegration Program, "ALP-AIR," Grant 334084), and partly by the Austrian National Science Fund (FWF) grants P30600 and P 33701. L.K received funding through the University of Innsbruck. S.J. was supported by the project SustES - Adaptation strategies for sustainable ecosystem services and food security under adverse environmental conditions (CZ.02.1.01/0.0/0.0/16_019/0000797). We used atmospheric data from the Innsbruck/University TAWES station, provided by the Austrian Weather Service ZAMG and the Department of Atmospheric and Cryospheric Sciences, Universitat Innsbruck. The City of Innsbruck is acknowledged for making the city tree inventory available, Michael Steiner for complementing it with trees in private spaces.

## Conflict of Interest

There is no conflict of interest.

## Code and Data availability

The eddy covariance flux code used to analyze fluxes can be accessed via Github (https://git.uibk.ac.at/acinn/apc/innflux). Data can be shared upon request.

## Author contributions

LK and TK designed and conceived the manuscript. MG was leading the instrumental operation of the PTR-TOF-MS for the 2015 and 2018 campaigns. MG, SJ, AP and MS performed the raw data processing of NMVOC data. MS, TK and MG performed the NMVOC flux analysis. GW provided input on tree species information. TK and LK performed analysis regarding BVOC emission modeling. SJ aided in the operation of the PTRTOFMS and raw data processing of NMVOC data for the 2018 campaign. All authors provided input and contributed to writing the manuscript.

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

**Table 1. Literature values of the 44 most abundant tree species found in the 1 km² area surrounding the measurement site. All**
**values are given in mg g(dry weight)$^{-1}$ h$^{-1}$. Reference subscripts refer to a) Stewart et al., 2003, b) Kesselmeier & Staudt 1999, c)**
**Karl et al. 2009, d) Noe et al. 2009, e) Nowak et al. 2002, f) Wang et al. 2007, g) Baghi et al 2012, h) Li et al 2009, i) Owen et al.**
**2003, *) standard value of 0.1, as no literature value available.**

| plant species name | number of trees | ISO emission potential | MT emission potential | SQT emission potential |
|---|---|---|---|---|
| *Acer platanoides* | 202 | 0.02[a] | 1.83 [a] | 0.1[c] |
| *Betula pendula* | 151 | 0.05[a] | 2.80 [a] | 2[c] |
| *Aesculus hippocastanum* | 98 | 0.10[a] | 0.10 [a] | 0.1[*] |
| *Fagus sylvatica* | 97 | 0.01[a] | 0.36 [a] | 0.1[c] |
| *Fraxinus excelsior* | 90 | 0.00[a] | 0.00 [a] | 0.1[c] |
| *Prunus avium* | 85 | 0.10[a] | 0.24[a] | 0.1[c] |
| *Robinia pseudoacacia* | 85 | 11.87[b),c),d] | 2.48 [b),c),d] | 0.1[c] |
| *Acer pseudoplatanus* | 77 | 0.00[a] | 0.00[a] | 0.1[c] |
| *Picea abies* | 68 | 1.07[a] | 4.00[a] | 0.1[c] |
| *Pinus sylvestris* | 68 | 0.10[a] | 6.45[a] | 0.1[c] |
| *Tilia platyphyllos* | 54 | 5.50[a] | 0.10[a] | 0.1[c] |
| *Taxus baccata* | 52 | 0.10[a] | 0.10[a] | 0.1[*] |
| *Cornus mas* | 40 | 0.10[e] | 1.60[e] | 0.1[*] |
| *Populus alba* | 40 | 53.00[a] | 2.30[a] | 0.1[c] |
| *Prunus cerasifera* | 37 | 0.10[a] | 0.79[a] | 0.1[*] |
| *Quercus robur* | 36 | 38.45[a] | 0.94[a] | 0.1[c] |
| *Populus nigra* | 35 | 52.50[a] | 2.30[a] | 0.1[c] |
| *Cupressus sp* | 32 | 0.10[a] | 0.90[a] | 0.1[c] |
| *Carpinus betulus* | 30 | 0.10[a] | 0.04[a] | 0.1[c] |
| *Acer campestre* | 29 | 0.05[a] | 0.10[a] | 0.1[c] |
| *Salix alba* | 24 | 37.20[a] | 1.10[a] | 0.1[c] |
| *Platanus acerifolia* | 22 | 20.00[a] | 0.05[a] | 0.1[*] |
| *Tilia cordata* | 21 | 0.00[a] | 0.00[a] | 0.1[c] |
| *Prunus serrulata* | 18 | 0.10[a] | 0.79[a] | 0.1[*] |
| *Acer saccharinum* | 17 | 0.10[b] | 2.85[b),c] | 0.1[*] |

| *Cupressus sempervirens* | 15 | 0.00[c] | 0.70[c] | 0.1[c] |
|---|---|---|---|---|
| *Abies alba* | 14 | 1.00[c] | 1.50[c] | 0.1[c] |
| *Pinus cembra* | 14 | 0.00[c] | 2.50[c] | 0.1[c] |
| *Sophora japonica* | 14 | 10.00[f] | 0.10[f] | 0.025[*] |
| *Thuja occidentalis* | 14 | 0.00[c] | 0.60[c] | 0.025[c] |
| *Ginkgo biloba* | 11 | 0.30[h] | 0.60[h] | 0.025[*] |
| *Malus domestica* | 11 | 0.50[a] | 0.60[a] | 0.025[c] |
| *Sorbus aucuparia* | 11 | 0.50[a] | 0.10[a] | 0.025[c] |
| *Gleditsia triacanthos* | 10 | 0.10[e] | 0.70[g],[e] | 0.025[*] |
| *Sorbus intermedia* | 10 | 0.50[a] | 3.00[a] | 0.025[*] |
| *Aesculus carnea* | 8 | 0.00[g] | 12.00[g] | 0.025[*] |
| *Chamaecyparis lawsoniana* | 8 | 0.10[i] | 0.67[i] | 0.025[*] |
| *Liquidambar styraciflua* | 8 | 46.58[b] | 19.17[b] | 0.025[*] |
| *Magnolia Kobus* | 8 | 0.05[a] | 3.25[a] | 0.025[*] |
| *Platanus hispanica* | 8 | 20.00[a] | 0.05[a] | 0.025[*] |
| *Acer palmatum* | 7 | 0.05[a] | 1.83[a] | 0.025[*] |
| *Juglans regia* | 7 | 0.00[b] | 1.40[b],[c] | 0.025[c] |
| *Larix decidua* | 7 | 0.00[c] | 5.00[c] | 0.025[c] |
| *Platanus occidentalis* | 7 | 20.00[a] | 0.05[a] | 0.025[c] |
