# Peer review of "Interannual Variability of Terpenoid Emissions in an Alpine City"

_Atmospheric Chemistry and Physics, 2021_

## Author Comment (AC1)

**Reviewer 1:**

**General Comment:** The authors have attempted to delve deeper than a mere presentation of measured fluxes and correlation between variables. For example, they estimate emissions of the isoprenoids within the footprint of the flux tower using standard emissions algorithms and a detailed tree inventory, and attempt to attribute the anomalously high standardised isoprene emissions observed in 2018. However, their description of the methods used, and their interpretation and discussion of their results are at times surprisingly superficial, detracting from the important addition to real-world observational data and knowledge this paper should have provided.

Before this paper is accepted for publication in ACP, the authors should supply far greater detail of the measurements made and the calculations and analyses performed, rather than entirely relying on previous publications. The purpose of the methods section is to supply the reader with sufficient detail to fully understand what has been done, how and why; further detail required to enable replication of the study can be left to the previous studies conducted at this location.

*Reply: We thank reviewer 1 for the valuable comments and assessment. We have deepened the analysis and discussion of the results where possible. Given the exploratory nature of this study we acknowledge the fact that we can not address every single aspect that might arise from the presented observations. Some limitations have to do with the fact that we refrain from overinterpretation of several ancillary data that we used to qualitatively support some of the hypothesis (e.g. spatial analysis of remote sensing products for soil moisture). Others have to do with the fact that important variables such as mean root depth etc. are simply not known. We have clarified this in our detailed response (see below). In addition we expanded our discussion on the methodology and analysis steps as requested.*

**Comment:** 2.1 Field site and instruments:

Please give details of the sampling frequency of the CPEC200, and any periods for which flux data are unavailable.

*Reply: The sampling frequency was 10Hz. In 2015 there was a data gap between July 31 10:45 and August 03 11:15. There were no data gaps in 2018.*

*Change: We added the sampling frequency and mentioned the data gap in 2015 in the text.*

**Comment:** Likewise the PTR, and explain the significance of the acetone and isoprene sensitivities. Was the PTR operated in full mass scan or selective scan mode? How long was the inlet line from the tower to the PTR? What are the estimated wall and chemical losses of VOCs, particularly the sesquiterpenes and lower volatility and more reactive monoterpenes, in the inlet tube?

*Reply: The PTR instrument used in this study is a PTR-QiTOF-MS. A time-of-flight mass-spectrometer inherently provides full mass scans. At 40 µs TOF extraction period, 2500 arrival*

*time histograms are co-added and processed to full mass-spectral information at 10 Hz. Frequent zero and span calibrations (here automatically performed several times per day) provide detailed information on sensitivity, limit of detection and general performance. In PTR-MS literature (see for instance review by Yuan et al. 2017; http://dx.doi.org/10.1021/acs.chemrev.7b00325) compound sensitivities of acetone (representative of oxygenated VOC) are frequently reported along with those of NMHCs of which isoprene is representative. Sensitivities reported here (1550 Hz ppb$^{-1}$ and 950 Hz ppb$^{-1}$, respectively) are favorable compared to those in the literature (see Yuan et al. 2017). High sensitivities result in low flux detection limits. Isoprene sensitivities in the order of 60% of those of acetone are consistent with theoretical reaction rate coefficients for proton transfer with hydronium ions (see Cappellin et al. 2012; https://pubs.acs.org/doi/10.1021/es203985t). The high mass resolving power of the instrument allows for the separation of isomeric VOC species (Graus et al. 2010; http://dx.doi.org/10.1016/j.jasms.2010.02.006) .*

*The inlet line (Teflon PFA, ¼" ID x 12.7m heated at 30°C) is purged at a flow rate of ~19 slpm (STP) via a pressure controller resulting in a plug stop residence time of 0.4 s at a pressure of 780 hPa.*

*Cospectral analysis (now added as Supplementary Information) shows the expected -7/3 power law behavior in the inertial subrange for isoprene, monoterpenes, and sesquiterpenes. Loss of high frequency response due to wall interaction would result in a steeper tailing off towards high frequencies. Since this is not the case, wall loss of volatile isoprenoids is expected to be negligible.*

*Change: Text was clarified by adding information on the performance of the PTR-QiTOF-MS and the EC inlet system.*

**Comment:** How reliable a measure of precipitation at the urban site is the tipping bucket gauge? Presumably this has been evaluated in previous studies. Likewise, the SMAP retrievals: at 9km resolution, how well do they capture the fine detail of heterogeneity in surface across an urban area? Also, it is not clear the purpose of the SMAP retrievals as the authors barely make reference to them and do not appear to make use of these data in their estimates of isoprenoid emissions or discussions of shortcomings in their study and areas of future research.

*Reply: The precipitation measurements are operated by Zentralanstalt für Meteorologie und Geodynamik (ZAMG, Austrian Met-Service) at the station Innsbruck Universität (WMO SYNOP number 11320). The station has been in operation since 1877.*
*The main purpose of the SMAP soil moisture retrievals is to reinforce that the lower precipitation in 2018 resulted in less soil moisture being available to plants. Due to the large footprint of the SMAP product, it is not the aim to use these data in any further analyses, e.g. in the MEGAN model.*

*Change: Details regarding precipitation measurements at the official weather station along with its operator (ZAMG) were added to the text. The motivation for including the SMAP has been clarified in the revised manuscript.*

**Comment:** 2.2 Eddy covariance fluxes

Why was the dataset reduced to daytime hours (although this is not apparent from the presented plots of diurnal profiles of fluxes)? Both monoterpene and sesquiterpene emissions are predominantly temperature controlled and hence continue throughout the night. Isoprene accumulation overnight has been reported on numerous occasions, with Millet et al (2016; doi: 10.1021/acs.est.5b06367) attributing an early morning burst of ozone formation in an urban area to isoprene emissions late the previous evening. It is more usual for nighttime fluxes to be filtered out by too low windspeed if and when a stable nocturnal boundary layer is established. Why have the authors not simply followed this established methodology?

*Reply:  Here we focus on process level understanding of underlying BVOC emission relationships and see how well these can explain our observations. We do not attempt to look at total BVOC budgets of biogenic NMVOCs by including often uncertain nighttime data as the reviewer points out. Neither do we speculate about nighttime and previous day chemistry. Due to the change of footprint during the day, this study focuses on daytime fluxes of these BVOCs in the east sector. We would disagree that there is the established method to investigate BVOC emissions from different ecosystems. For example in urban areas more emphasis has to be taken on flux footprints. The above referenced paper describes advection of isoprene enriched air masses over a city downwind of a forest, thus a quite different scenario to what we observe here. Isoprene concentrations in Innsbruck for example do not accumulate at night as reported in the reference by Millet et al. 2016. The focus of the current study is also not on isoprene mixing ratios and consequences on air chemistry. Because isoprene production is largely driven by light, and we want to compare isoprene fluxes to monoterpene and sesquiterpene fluxes in a consistent fashion, we focus on daytime flux patterns.*

*Change: We clarified the corresponding section, added new plots on flux density footprints and explain in detail the underlying rational used to investigate BVOC fluxes here.*

**Comment:** How many measurements were excluded? What proportion of measurements were suitable for flux calculation and subsequent analysis in each year? Please explain more clearly the filter that was applied (L142-146).

*Reply: 29% of data in 2015 and 11% of data in 2018 had to be excluded as they did not meet the QA/QC criteria (horizontal inflow sector for CSAT3 criterium (see Chapter 4.2.5. in Foken (2017)) causing the difference between the two periods).     In 2015 the tower was at a provisional location with a sonic anemometer heading direction of 76° and flow distortions for westerly winds due to the building and support structure; in the course of the establishment of the IAO lab the sonic anemometer and inlets were moved ~50m to the southern edge of the building with a heading of 129°. The wind sector to be excluded as a consequence of potential flow distortions now is (309+/-30)°. With main wind directions of the valley wind system being 65° (up-valley flows out of the NE sector) and 235° (down-valley flows out of the SW sector), since 2017 data need to be excluded less frequently.*

*For comparability isoprenoid fluxes in this study are therefore limited to the northeastern sector of [0°,120°] in both years.*

*Change: A sentence explaining the amount of reduction of data each year by implementing the QA/QC criteria was added to the updated manuscript.*
*The argument for restricting the analysis of isoprenoid fluxes to the sector [0°,120°] was consolidated in 2.1.*

**Comment:** While the footprint of the flux tower is shown in Fig 1, it would be more instructive to see the footprint density, which would benefit from a fuller explanation in the text. I assume that by density, the authors are referring to an estimation of the relative contribution of each point within the footprint to the air mass samples at the flux tower. How is the contribution determined? What weighting system is used? Simply by air mass or by proportion of $CO_2$ and $H_2O$ flux?

*Reply: As the reviewer suggested we changed figure 1 to show different isolines of the footprint density for both years separately. The footprint density shows relative contributions of air mass samples at the flux tower.*

*Change: We changed Figure 1 to show footprint density isolines for both years.*

**Comment:** The authors need to explain why the 2018 footprint is so much smaller than the 2015 footprint. To me that suggests that windspeeds were considerably lower in 2018 - yet the authors demonstrate that AVERAGE windspeeds were similar. I think Fig 1 would benefit from the inclusion of the a panel showing the footprint density and windrose plots for each year - this would considerably help understanding of these issues.

*Reply: To the reviewers previous comment we changed Figure 1 to show footprint density isolines. The new figure 1 shows strong similarities of the footprint in the NE direction, the predominant wind direction during the daytime. As in the previous version of the manuscript all data analysis is restricted to this NE wind sector. Differences in the other wind directions potentially stem from a slightly different flux tower location.*

*Change: We changed Figure 1 to show footprint density isolines for both years.*

**Comment:** Why did the authors choose to use MEGANv2.0 to calculate isoprene emissions AND include only the light and temperature activity factors? The biggest limiting factor to photosynthesis and therefore availability of electrons and carbon for isoprenoid synthesis is water availability, which the authors have via the SMAP retrievals. Multiple studies have demonstrated the importance of accounting for soil moisture status (in addition to the observational studies the authors cite, Sinderalova et al (2014; doi:10.5194/acp-14-9317-2014), Emmerson et al (2019; doi: 10.1016/j.atmosenv.2019.04.038), Otu-Larbi et al (2021; doi: 10.1111/gcb.14963) have applied various models to show this, and Jiang et al (2018; doi: 0.1016/j.atmosenv.2018.01.026) have presented a new parameterisation of soil moisture impacts on isoprene emissions specifically for MEGAN). I can understand that the authors may wish to

start with the "standard" algorithms, but why not use this in the subsequent exploration of "other factors" that may account for the 2018 anomaly?

*Reply: The short term variations explain most of the variance for such a dataset and are used to factor in different temperature and light regimes between the two years. Due to the large footprint of the SMAP soil moisture product, we felt that the use of these data to constrain MEGAN was not warranted. In section 3.3 we do go through a detailed discussion of our results including an in-depth discussion about other factors. The main challenge for including a quantitative analysis of drought effects is that additionally a representative soil moisture depth profile in the urban area would be necessary. This is currently not available - as such the Megan 3 parameterization that is for example used in the CLM4.5 global model, would not fully capture local conditions - the question would then be what insights can be gained by randomly tuning soil moisture parameters to fit the observations.*

*Change: The motivation behind the use of the SMAP soil moisture product has been added to the revised manuscript.*

**Comment:** Monoterpene and sesquiterpene emissions are very definitely NOT "known to be purely temperature dependent"! As early as 1995, Staudt & Seufer (Naturwissenschaften 82: 89–92) reported light-dependent emissions of monoterpenes and this is now well-accepted AND, importantly, is explicitly included in the emissions algorithms for mono- and sesqui-terpenes in MEGAN2.1 (Guenther et al, 2012; doi: 10.5194/gmd-5-1471-2012). Why have the authors not used these more recent formulations? (Even if this serves to support their later statement that at this location fluxes are solely temperature dependent)?

*Reply: We acknowledge that monoterpene emissions can also be light dependent and added the proposed references. We agree that the word 'purely' in this context is not correct. In our manuscript we do not claim that monoterpenes are exclusively temperature dependent. The reviewer actually also acknowledges this in a previous comment (ie. see comments above: "Both monoterpene and sesquiterpene emissions are predominantly temperature controlled and hence continue throughout the night..."). For sesquiterpene emissions the situation of light dependent emissions in the literature is less clear, Guenther et al., 2012 assumes 0.5 across all SQTs. We follow the reviewers comment and modify the emission parameterizations in a revised figure 2.*

*Change: We corrected the wording in section 2.2 on temperature dependent terpene emissions, added corresponding references and changed the flux vs. algorithm comparison in Figure 2 to Guenther 2012.*

**Comment:** How were the effects of LAI and leaf age on emissions accounted for?

*Reply: No direct LAI measurements are available for this study. Both campaigns were conducted during a similar time of year which should lead to comparable leaf age. City gardeners confirmed that there was no early senescence in either of the two years.*

*Change: We added a sentence to the text on our assumptions of LAI as well as that we do not have such observations available.*

**Comment:** The authors state that the chemical lifetime of sesquiterpenes against oxidation primarily via ozonolysis is of a similar order of magnitude as turbulence timescales. The same will be the case for the more reactive monoterpenes, many of which could be expected to be emitted from the mix of tree species shown in Table 1. Why have the authors not also accounted for the chemical loss of a proportion of the monoterpenes? Also, while the authors have given an equation for the chemical loss, is not at all clear how and when this was applied to their estimation of sesquiterpene emissions and fluxes. Please elucidate. For example, the chemical loss rate will be highly dependent on the availability of atmospheric oxidants, particularly ozone in this case, but also the nitrate radical at night, and the concentration of the isoprenoid, What assumptions have the authors made in this regard, and how is this justified?

*Reply: We now provide a more complete description of the calculation of the loss of terpenes including estimates for monoterpenes. Since we do not have speciated terpene fluxes, we performed a sensitivity study (e.g. estimating realistic bounds) assuming a fraction of the total sesquiterpene (or monoterpene) flux was composed by the most reactive compound (rSQT and rMT). For sesquiterpenes, for example, we can take the estimated rate constant for ozone and beta caryophyllene: 1.2e-14 cm3/molecules/s. A typical compositional mix of sesquiterpenes was reported by Sakulyanontvittaya et al., 2008, https://pubs.acs.org/doi/pdf/10.1021/es702274e), who assessed reactive terpene fractions between 36-50%. Typical reaction rates of less reactive sesquiterpenes (nrSQT) (e.g. cedrene, longifolene: Atkinson et al., 1994: doi: 10.1002/kin.550261207) are on the order of 1 to 10 x 1e-17 cm3/molecules/s. Taking these boundary conditions gives a realistic range of the reacted fraction of measured SQT fluxes. Similarly we can do the analysis for monoterpenes, where the fraction of reactive terpenes (rMT) such as ocimene is typically lower (e.g. 10 - 15% - Sakulyanontvittaya et al., 2008,). For comparison, trans-beta-ocimene, one of the most reactive monoterpenes known to be emitted from plants, has a reaction rate constant of 2.6e-14 cm3/molecules/s. Figure S2 and S3 in the supplement show the non-reacted flux for total sesquiterpenes due to reaction with ozone assuming a 36 to 64 and a 50 to 50 mix (rSQT to nrSQT). With these scenarios daytime reductions of total sesquiterpenes fluxes due to chemistry would be on the order of 30-45%. For monoterpene fluxes we calculate losses on the order of 12% (Figure S4).*

[Figure]

*the same plot for a 50% to 50% fraction is shown in the following figure*

[Figure]

*The same analysis for the loss of reactive monoterpenes with a 12% ocimene contribution to the overall MT mix ( https://pubs.acs.org/doi/pdf/10.1021/es702274e) is shown in the following figure:*

[Figure]

*Change: We have clarified the requested issues on chemical losses on measured fluxes in the revised paper, and added an in depth discussion on the performed sensitivity calculations.*

**Comment:** 2.3 City tree inventory

What % of the trees are contained in private gardens? And how many trees would the authors estimate are unaccounted for in the inventory.

*Reply: 59 % of the trees in the study area are found in private gardens and we estimate our tree inventory to cover > 90 % of all trees in the study area*

*Change: The information requested by the reviewer has been added to the revised manuscript.*
**Comment:** 2.4 Emission potentials

Why did the authors not use the emission potentials, or parameterisations for calculating them, from Guenther et al (2012; doi: 10.5194/gmd-5-1471-2012)?

*Reply: We changed the comparison between measured fluxes and emission parameterization in Figure 2 to the newer algorithm of Guenther et al 2012. For Figure 4 we think it is best to work with the more specific emission potentials from the individual tree types rather than a*

*more general emission potential from a global model as it would not be clear which plan functional type to use.*

*Change: Figure 2 changes as described above, no changes to Figure 4.*

**Comment:** 2.5 Relative emission ratio maps

Why choose such a coarse grid as 100m x 100m when a full tree inventory is available for the footprint of the tower (which was <1km x1km each year)?

*Reply: The 100m x 100m grid was chosen to achieve higher statistical value. Footprint density is not part of the calculation of the relative emission map.*

*Change: We added a count of trees per grid to Figure 5 to show this statistical value. We clarified the text that footprint density was not used as part of calculation of the map in Figure 5*

**Comment:** What effect would the authors expect changes in leaf age and phenology, and differences between different tree species, to have on the estimated ratios? This does not appear to be discussed anywhere.

*Reply: We acknowledge this comment but have no data available to calculate these different effects on the estimated ratios.*

*Change: We added a sentence explicitly stating that we don't have leaf age and phenology data available and an estimate of the robustness of the analysis (see below).*

**Comment:** 3.1 Flux footprint

Again, the authors refer to the footprint density but do not explain how this is derived (see previous comment).

*Reply: We appreciate this comment and have addressed this issue on the similar comment above.*

*Change: See above*

**Comment:** While the authors do point out that they are unable to calculate absolute values of emissions as they do not have data for the leaf dry weight for the different trees and so focus on the ratios of the different terpenoids, I would expect to see further analysis of discussion of the robustness of this approach. For example, in this first paragraph, they discuss the contributions of the different tree species to absolute values of each isoprenoid without consideration of differences in total leaf mass of the different species, nor the potential for differences in leaf mass between years or over the course of the measurement period which extends into a time when some of the trees are likely starting to senesce.

*Reply: We appreciate this comment. As it would go beyond this study to analyze all these uncertainties of unmeasured parameters in detail we performed a perturbation analysis changing the assigned emission potential of the 20 highest emitters in the area assigning them double and half of their emission potential. This should give a first estimate on the robustness of this ratio approach. For the average emission ratios over the study area this resulted changes on the order of 5-15%.*

*Change: We added a sentence describing the perturbation analysis and results of ratio robustness to the manuscript.*

**Comment:** How do the authors account for the strong horizontal heterogeneity of the flux footprint? A fundamental assumption required for applying EC techniques is that the fetch is homogeneous in all directions from the flux tower. This is very definitely  not the case here. I would suggest that at the very least, they should split out the fluxes and bottom-up emissions by wind direction - this would provide a far more powerful analysis of the drivers of the potential emissions and observed fluxes.

*Reply: This is exactly what we did by constraining our fluxes to the east sector. It is a wrong assumption that the fetch has to be homogeneous in all directions (this would be an ideal case, where you then would not need to worry much about flux footprints). Also, why just singling out the EC technique here - in fact the EC technique is more robust than for example the gradient technique under these conditions! The heterogeneity for EC data analysis is implicitly accounted for by the stationarity criterion, that the data were filtered for. Also, during daytime advection terms due to heterogeneity are minimized.*

*Change: No change.*

**Comment:** 3.2 Two summers of urban BVOC fluxes

The authors appear to use "flux" and "emissions" interchangeably throughout this and subsequent sections. The two are NOT equivalent and the authors should be explicit about this. While the fluxes can be taken as a good indicator of the pattern and magnitude of emissions in the footprint, they are not measuring leaf- or tree-scale emissions, particularly of the more reactive species.

*Reply: We acknowledge this comment and ensured that the word flux is used whenever we talk about measured eddy-covariance data and the word emissions for leaf- or tree-scale emissions*

*Change: Inconsistencies of the word uses were corrected.*

**Comment:** Similarly, "footprint" and "footprint density".

*Reply: We now show the footprint density and state that this pertains to flux footprints.*

*Change: We cleaned up the word use.*

**Comment:** Presumably the authors use the winter-time benzene/isoprene ratio in an attempt to exclude biogenic sources of isoprene. However, I would expect the sources of anthropogenic VOCs to differ between summer and winter, resulting in differences in magnitude but also ratios of different VOCs between seasons. It's not clear how the authors extrapolated from the winter-time ratio to deduce that 70-80% of the isoprene measured at the flux tower during the campaign was biogenic in origin.

*Reply:* **In an urban setting the most important anthropogenic sources for these compounds are traffic (e.g. Borbon et al., 2001; doi: 10.1016/S1352-2310(01)00170-4). The analysis presented here is based on flux ratios, which are more closely related to emission ratios, and not concentration ratios. It is not clear why the reviewer expects substantially different emission ratios from traffic between isoprene and benzene between seasons. Tailpipe isoprene emissions (like 1,3 butadiene or benzene) are primarily related to combustion processes. Unlike toluene there is little or no evidence of major evaporative losses of isoprene. To test this hypothesis we calculated emission ratios between 1,3 butadiene (a useful surrogate for isoprene) and benzene using COPERT. The ratio between 1,3 butadiene / benzene for hot+cold vs hot+cold+evaporative varies by about 2-5%. This is also corroborated by a study published by Borbon et al., 2001 (doi: 10.1016/S1352-2310(01)00170-4) showing little temperature dependence of the isoprene to benzene emission ratio from traffic. In fact as demonstrated by many studies (e.g. Borbon et al., 2001 and others), the major influence of seasonal variation of isoprene emissions stems for biogenic sources and not anthropogenic emission sources. The emission flux ratio between isoprene and benzene obtained in the non-growing season should therefore be a reasonable approximation for the traffic related contribution dominating anthropogenic emissions. For this analysis we assume that the local biogenic contribution to urban benzene emissions is negligible. A similar approach based on concentration variations has been used in the past to differentiate biogenic vs anthropogenic isoprene emissions (e.g Reimann et al., 2000: doi: 10.1016/S1352-2310(99)00285-X), who calculated that the ratio between isoprene and 1,3 butadiene was 0.42. Taking these values in combination with COPERT derived 1,3 butadiene and benzene emission factors, we can calculate a potential tailpipe isoprene emission that is 10% of that of benzene. It results in a daytime contribution of approx. 5%, and a nighttime contribution of at most 20%.**

[Figure]

*Instead of just referring to the worst case, we now state that the used methodology in more detail. our previous conclusion that isoprene emissions are dominated by biogenic sources remains.*

*Change: We included the above discussion and references in a revised manuscript.*

**Comment:** While the authors highlight the discrepancies between predicted emissions and measured fluxes for lower temperature for mono- and sesqui-terpenes, particularly during 2018, they do not similarly highlight the over-estimation of isoprene emissions for these same temperatures in 2018. In fact, they state that measured isoprene fluxes closely followed estimated isoprene emissions. Furthermore, for the mono- and sesqui-terpenes, I would expect to see a more detailed discussion of why this might be the case, rather than the brief statement that perhaps other vegetation contributed. This is undoubtedly true: mono- and sesqui-terpene emissions from grasses and herbaceous plants such as those likely commonplace in urban areas can be expected to be high. Why do the authors not, at the very least, consider the proportion of the footprint density covered by short vegetation (shown in Fig 1) and attempt to estimate what proportion of the flux may be accounted for by this?

*Reply: Within the overall uncertainty isoprene is represented well by the emission parameterization fit (2015: R2=0.64, 2018: R2=0.59 – now added to Figure 2) . For MT and SQT we find that a portion of the dataset contains flux measurements that are actually higher*

*than expected, and we attribute this to anthropogenic emission enhancements as has been shown in other studies (e.g. Gkatzelis et al. 2020. doi: 10.1021/acs.est.0c05471). We have stated this in the manuscript previously as the most likely explanation (ie. see line 243-244). We have tested this hypothesis, by considering footprint variations and relative distributions between grasses and trees, which were minor. Variations in flux footprint and a relative distribution with higher grassland MT emissions as suggested by the reviewer can be excluded as an explanation.  Instead, the residual of non-explained MT and SQT fluxes correlates with aromatic fluxes. The following figure serves as an example for 2015 data. Plotted is the residual MT flux (predicted - observed) vs. benzene flux. There is a clear positive correlation with R2 ~0.75 (RMSE: 0.006204). We therefore argue that this is the most likely explanation for MT and SQT flux enhancements, not being reproduced by biogenic emission parameterizations.*

[Figure]

*Change: We added the above discussion in a revised manuscript in more detail.*

**Comment:** It's not entirely clear why the authors spend so much time comparing their standardized emission potentials to those measured at other urban sites, without a deeper analysis of the similarities and differences between the various studies. Why not simply give the range of previous fluxes and show that these are of a similar magnitude?

*Reply: We believe this discussion warrants a paragraph to put our results in context of other urban environments, which tend to be more complex in many ways. In our opinion we DO compare similarities and differences between different urban sites to the extent possible (e.g.*

*comparing landcover differences and flux magnitudes). It is beyond the scope of the manuscript to aim at modeling other urban sites, without having access to these datasets as the reviewer seems to imply.*

*Change: No change.*

**Comment:** In their analysis and discussion of sesquiterpene fluxes, the authors refer back to their "correction factor" which they state puts an upper limit on actual fluxes up to 2.5 times those measured. Please see my previous comments regarding the calculation, application and appropriateness of this factor. In particular, neither turbulent nor chemical timescales remain constant over the course of a day or the 6-week plus observation period, and given the differences in temperatures, windspeeds (and possibly directions) between years, the inter annual variation is likely substantial. Have the authors considered these factors? In particular, the chemical climate of the urban atmosphere through the measurement periods should be carefully considered and discussed. What assumptions are the authors making with regard to the oxidant budget, reaction rates and turbulence?

*Reply: Yes we have considered these factors, particularly changing turbulence and ozone concentrations. We agree with the reviewer that this merits a more in depth analysis as provided in our reply to previous comments. We have therefore taken the reviewer's suggestion and provide a more in-depth explanation of calculated reactive losses for terpene (SQT and MT) fluxes. We calculated the chemical loss based on the eq. in section 2.2 (flux loss = 1-exp(-tau_t/tau_c), and applied it to all half hour periods. Indeed our major findings won't change. The turbulent timescales can be calculated quite straight forward, in addition ozone data were obtained from a nearby air quality station. For the reacted terpene fraction we take typical bulk composition data observed for different land use types (e.g. Sakulyanontvittaya et al., 2008).*

*Change: We added more discussion as already indicated in our reply to a very similar comment (see above).*

**Comment:** It would be useful for the authors to present a measure of the goodness-of-fit between the observed and theoretical fluxes for each of the isoprenoids for each year. Does it vary by wind direction and/or speed?

*Reply: We added the R2 for each isoprene for each year to figure 2 the analysis is restricted to the NE wind sector as explained above.*

*Change: Updated figure 2 with R2 values.*

**Comment:** 3.3 Isoprene flux anomaly

The details of the temperature and light dependence (including 24h and 240h) should already have been fully introduced in the methods section, not here in the results. Furthermore, in L152 the authors state they are using the MEGAN 5-layer canopy approach but here in L285 they state they are using the big leaf approach. Which is it? They are very different in their formulation and

capability. How appropriate is either canopy (5-layer or big leaf both assume horizontal homogeneity and a relatively uniform vertical structure) for modelling an urban canopy?

*Reply: We clarify the used approaches in more detail. For most of the analysis we used a big leaf model approach. In order to investigate the potential effect of using a more complex approach we performed sensitivity studies using the MEGAN 5 layer canopy model, which for example explicitly calculates leaf temperatures of sun and shade leaves based on an energy balance approach. We recognize that both modeling approaches have uncertainties in context of dispersed urban vegetation.*

*Change: We now describe the models and their usage in a revised methods section more clearly and point out various constraints of using big leaf and 5 layer canopy modeling approaches in the context of urban studies.*

**Comment:** Figure 3 is barely referred to from the text yet it makes a critical point that the authors then go on to discuss in some depth. Far more analysis and insight is required here. Figure 3B shows that the anomaly (roughly) increases with temperature and PAR, not PAR alone.
It would be very useful to see the number of data points per T-PAR bin in Figure 3B. The authors should also take their analysis deeper and attempt to elucidate what other factors and conditions lead to the observed anomalies. Wind direction (and therefore synoptic-scale met conditions), wind speed, soil moisture, VPD, etc.

*Reply: The number of datapoints are now mentioned. Possible explanations of the anomalies were discussed in detail in section 3.3. We have commented on the difficulty of using SMAP data to quantitatively interpret soil moisture in a previous reply. It is not clear why synoptic scale meteorology should play a role for the interpretation of eddy covariance fluxes. We are discussing eddy covariance flux measurements and not concentration data. These eddy covariance fluxes are interpreted in context of a flux footprint, which by definition includes information on wind direction and wind speed!*

*Change: We have added the suggested statistics on T-PAR analysis.*

**Comment:** It was good to see that the authors attempted to find alternative explanations for the substantial anomalies in isoprene fluxes during 2018.
(a) While average wind speeds are relatively similar between the years, the median Obukhov lengths are very different (by nearly a factor of 2!). Please could the authors explain this difference and provide some insight into the likely effect on fluxes measured at the flux tower. Again, it should be noted that mono- and sesqui-terpenes are considerably more reactive than isoprene and do not directly inform whether changes in isoprene emissions should be expected.

*Reply: The measurement period in 2018 (affected by a heat wave) saw higher median sensible heat flux, $\overline{w'T'}$, (more convective situation) and higher median friction velocity, $u^*$, compared to 2015. The Obukhov length, L, is inversely proportional to sensible heat flux, but $u^*$ enters the calculation of L to the power of three, resulting in an increase of L in 2018 compared to 2015. Testing the FFP model (Kljun et al. 2015 doi.org/10.5194/gmd-8-3695-2015) for its*

*sensitivities to the input parameters (L and u\*) it turns out that u\* affects the flux footprint density function more than, L, where a factor of 2 difference has very little influence in the parameter sub-domain relevant for the summer studies. We changed the text in the manuscript accordingly. For the comparison of turbulence parameters between the two years the data should also be restricted to the NE sector, daytime period, and QA/QC level of six or less. The newly calculated statistics of parameters for sonic temperature, sensible heat flux, friction velocity, wind speed, and Obukhov length are now clearly summarized in a table in the Supplemental Information and referred to in the main text.*

*Change: We created a new table (Table S1) in the Supplemental Information with clearly summarized statistics and details how stronger vertical mixing in 2018 is responsible for the smaller extent of flux footprint densities. We refer to it in the main text as follows:*
*"Median daytime wind speed and direction in 2018 (affected by a heat wave) are similar to those in 2015 (Table S1). Median sonic temperature, sensible heat flux and friction velocity (see Table S1) were higher in 2018 resulting in stronger turbulent vertical transport with mostly friction velocity being responsible for the differences of the flux footprint density function between 2015 and 2018 (Fig 1)."*

**Comment:** (b) It should be noted that LAI and leaf area density can vary for reasons other than pruning. For example, early senescence, difference in nutrient availability, herbivore or pathogen infestation, etc.

*Reply: Good point, however the observations by the city gardeners rule out these causes.*

*Change: The text was expanded to clarify that the causes for differences in LAI mentioned by the reviewer did not play a role.*

**Comment:** (c) It seems likely that the increased water fluxes in 2018 are due to surface evaporation from both the soil and the bare surfaces of the city if watering was increased during that summer. The similar SMAP retrievals for the 2 years further supports this. It should also be noted that, while useful, SMAP retrievals only provide moisture content of the top layers of soil and not the root zone which is critical for accessibility of water for trees. Is this why the authors appear to distinguish between soil water and soil moisture?

*Reply: As for the motivation behind using SMAP data, see comments above. The synonymous use of soil water and moisture is not intended and we have replaced the term soil water consistently with the term soil moisture.*

*Change: The term soil water was replaced with the term soil moisture throughout the manuscript.*

**Comment:** In fact, the authors are incorrect: mild drought has been shown from multiple measurement campaigns and modelling studies to INCREASE isoprene emissions (see previous reference list) and could in part explain the anomaly (Otu-Larbi et al, 2020 for example found emission potentials increased by a factor of 2.5 during mild drought in temperate deciduous forest). Otu-Larbi et al (2020) also reported an apparent burst of isoprene emissions on rewetting.

I suggest that the authors need to investigate more thoroughly whether the anomalies in isoprene fluxes occur at times of mild drought or in Reply to rewetting following drought conditions. I would recommend that the authors split (c) into two sections: one dealing with soil moisture, drought and rewetting and the second with the parameterisation and specifically the choice of Topt as these are quite distinct.

*Reply: We acknowledge the reviewers comments here, and would like to point out that this is exactly what we have stated in our original manuscript on line 348. Nevertheless the clear mechanism why some studies have observed a slight increase (or at least no significant decrease like for CO2) at the beginning of drought is still debated. Another likely explanation in this context could be differences in the mean rooting depth of isoprene emitters relative to soil moisture profiles. Information we do not have for this study, but we agree with the reviewer that this merits more discussion in the manuscript. In the cited reference (Otu-Larbi et al., 2020 doi: 10.1111/gcb.14963), we do not find a concrete proof of rewetting effects, since no direct isoprene emission or flux measurements were presented. The authors speculate that a combination of SWC, leaf temperature and rewetting emission bursts gave the best fit to their modeled ambient isoprene concentrations. The use of a 1-D chemistry model to interpret isoprene concentrations, which are subject to emission, chemistry and advection processes, is by its own merit a challenging task. Here the soil moisture effect (Jiang et al., 2018 doi: 0.1016/j.atmosenv.2018.01.026) between 2015 and 2018 would predict 20% lower emissions in 2018 when using SMAP soil moisture data. The cumulative precipitation for July, August and September 2015 was 340 mm, and 258 mm for 2018. When taking just the campaign duration, the cumulative precipitation was 269 mm in 2015 and 136 mm in 2018. To explain our results with the rewetting effect is more complex, since we can not clearly parse out an exclusively wet and dry period. Precipitation data during the 2018 heatwave show intermittent periods of thunderstorm activity throughout the campaign resulting in 136 mm of rain during the 2018 campaign. We do not have locally representative soil moisture data in the city. Looking at precipitation data, we do not see a statistically significant difference in the isoprene flux bias after rain events.*

*Change: We comment on the above issues in a revised manuscript and added the suggestion of extending the discussion concerning rewetting incl. references. Emission model parameterization differences were already split into a separate section (d), which is now also slightly revised according to the reviewers suggestions.*

**Comment:** It should be noted that both mono- and sesqui-terpene emissions are also controlled by stomatal conductance which could be expected to affect emission rates during drought periods (se e.g. Niinemets and Reichstein, 2003a & b; doi: 10.1029/2002JD002620 & 10.1029/2002JD002626).

*Reply: We appreciate this comment and added a sentence to the manuscript explaining that difference due to drought would be expected for mono- and sesquiterpene emissions but that such differences were not observed in our flux measurements between the two summers.*

*Change: We added a sentence as well as the mentioned citations.*

**Comment:** How accurately does the MEGAN canopy (again the authors refer to the 5-layer version here) represent LEAF as opposed to air temperature? This is usually anomalously high during periods of drought and other abiotic stresses (Niinemets, 2010; doi: 10.1016/j.tplants.2009.11.008 & Potosnak et al, 2014; doi: 10.1016/j.atmosenv.2013.11.055).

*Reply: We do not have individual (and representative) leaf level temperature measurements of city trees to test whether modeled leaf temperatures are too high or too low. It could be an explanation that we have previously also discussed, e.g. line 355cc, where we mentioned that temperature response curves have been observed to change under heat/drought stress. Since we focus on a comparative study between the years 2015 and 2018 (i.e. investigate what can explain the relative differences between the two years), it is important to keep in mind that we are not necessarily only interested in an absolute value of leaf temperature, rather we are interested to see whether a physically plausible difference in leaf temperatures between 2015 and 2018 can explain the relatively high fluxes during the heatwave in 2018.*

**Comment:** 3.4 Top-down and bottom-up BVOC flux ratios

Please see previous comments regarding the treatment of chemical loss of sesquiterpenes in the analysis of flux vs emission. To my mind, this is a substantial weakness of the authors' approach (assuming I have correctly understood how the correction factor is applied) as it appears to involve a gross and relatively unjustified assumption regarding the oxidative capacity of the urban atmosphere during the measurement periods.
The conclusion that more studies of sesquiterpene emission potentials are needed is weak. Much is still required to be understood about sesquiterpene synthesis, emission, dispersion and atmospheric reactions before fluxes and emissions can genuinely be compared.

*Reply: We restructured the discussion on reactive losses and added a more detailed analysis of potential chemical losses for the bulk flux analysis (see our response to earlier comments). It is important to mention that we considered the variation of turbulent mixing and variations of oxidants (e.g. ozone) in this analysis. Assuming bottom up speciation of SQT and MT we provide estimates of likely constraints on chemical losses on observed fluxes. We believe the reported SQT fluxes can be used to gain insight on emissions by considering the potential chemical losses, and have rephrased this section accordingly.*

*Change: We significantly expanded the section on how we constrained atmospheric losses on measured SQT and MT fluxes.*

**Comment:** Figure 5 is very poor - the colour scales are such that the panels provide very little useful information. Presumably the squares with apparently no overlay contain no vegetation at all, but why not white them out? Again, it would also be useful to see how many flux observations originated from each of the grid cells (this again comes back to the matter of the missing figure of footprint density).
*Reply: We thank the reviewer for this comment and we changed figure 5 accordingly (see below).*
*Change: Figure 5 color scale was improved and areas with no trees are colored in white. We added a fourth panel showing the number of trees in each square as well.*

**Comment:** 4 Summary

This section is rather superficial and the conclusions weak. In particular, the authors again refer to having ruled out the effect of severe drought, apparently unaware that in several of the studies they have cited, that isoprene emissions are substantially enhanced during periods of MILD drought. Possible causes of this are well discussed by both Potosnak et al (2014) and Ferraci et al (2020). Plus they have not attempted to analyse whether their observed anomalies coincided with officially recognised drought conditions in the city. Their study certainly does not show that urban conditions are distinct from other ecosystems. The unexpectedly high isoprene concentrations during heatwave-droughts have been reported from woodlands as well.
Again, the final statement that more work is needed is weak. Precisely what lab- and field-based experiments and modelling is required in the authors' opinion?

*Reply: The effect of isoprene enhancements during mild drought is still somewhat under debate. For example Geron et al., 2016, doi: 10.1016/j.chemosphere.2015.11.086, speculate that it has more to do with plant structural variations (root depth vs changing water table / soil moisture changes). Some level of drought conditions in the city were officially recognized as the city started watering city trees, something that is normally not done in this climate region. We concur with the reviewer that increased isoprene emissions during mild drought in 2018 could have played a role. We do not claim that the physiology and biochemistry of urban plants are necessarily different, but it is obvious that environmental parameters (e.g. air pollution, light environment) and land management (eg. watering) practices represent distinct differences compared to natural forests. The urban heat island is another well established example that represents a significant environmental difference to natural forests.*

*Change: The above has been clarified in a revised manuscript.*

**Technical comments:**
**Comment:** While mostly clearly written and presented, the manuscript would benefit from English editing to clarify some statements and explanations. I have suggested replacement text only where the original meaning is unclear.
*Reply: We thank the reviewer for these detailed technical comments and will in the following only mention the changes.*

Abstract:

L17 (and throughout document): Standard scientific notation should be used, i.e. $3.0 \times 10^{-3}$ rather than $3.0 . 10^{-3}$, etc.
*Change: All instances are changed*

**Comment:** L21 - please replace "explained" with "explain"
*Change: Replaced*

**Comment:** L21 - please replace "standard emission potentials" with "standardized isoprene emission potentials"

*Change: Replaced*

**Comment:** 2 Material and methods

L128-129 - please give the conversion used to estimate PAR from short wave radiation.
*Change: The relationship previously only cited was not added explicitly to the manuscript: PAR/SW radiation ~ 0.46 during summer, daytime conditions.*

**Comment:** L173 - What is meant by "overlapping"?
*Change: We clarified the text in the revised manuscript that with overlapping was meant that for each plant species in our tree inventory that was also measured by Stewart et al 2003 we used the emission potential from that study.*

**Comment:** L178 - "IS" should read "ISO"
*Change: Replaced*

**Comment:** L186-7 - This is a little unclear. I suggest that the authors demonstrate the full calculation specifically for ISOtile/MTtile.
*Change: This was clarified in the revised manuscript.*

**Comment:** L200, 205 & 208: The same 12, 19 & 38 trees in 2015 and 2018? (As the footprint does differ)

*Reply: Yes, these trees have dominant emission potential and the differences in footprint density are overpowered by the emission strength of these emitters. We improved figure 1 now showing more footprint isolines to show that the two years in footprint density are not as different as the previously shown 60% footprint density isoline indicated. The same number of trees don't account for 100% or the emissions but certain percentages as mentioned in the manuscript. The exact percentage of total emission is somewhat different in the two years, this is the effect of the differences in footprint density. The percentages are given in the manuscript. The exact order of percent influence also varies between the two years due to footprint density differences but overall a large percentage of emissions arrive from the same trees.*

*Change: No change.*

**Comment:** L213 - If the authors have discounted nighttime fluxes, it is not clear why they

consider nighttime emissions (which I assume they do as they refer here to differences in nighttime temperatures).

*Reply: Nighttime fluxes are presented in Figure 2 for the sake of completeness but not further analyzed. Nonetheless are night-time temperatures important as emissions have a long term temperature dependency. E.g. Guenther et al 2012 describe dependency terms of the past 24h and 240h average temperatures.*

*Change: No changes as the importance of past 24h & 240h temperature importance is described later in the manuscript in a more relevant section.*

**Comment:** Figure 1: The difference between dark green and light green for trees and short vegetation is not sufficiently distinct.

*Change: The colors where improved in the revised manuscript.*

**Comment:** L225 - I would suggest replacing "BVOC" with "isoprenoid" as the authors do not report the fluxes of any other BVOC.
*Change: Replaced*

**Comment:** L226-228 - Why do the authors refer back to parameterisations developed in 1993 and 1994 rather than the ones they have actually used? (And see previous comments regarding the light-dependence of monoterpene emissions).
*Change: We changed both Figure 2 and the text to refer to Guenther et al 2012.*

**Comment:** Figure 2A-C - Why have the authors presented a full diurnal cycle, when they explicitly state in the methods that they consider only "daytime" fluxes, which they further refine to 09:00-16:00 LT?
*Change: We revised Figure 2 and the text to clarify that we use the full diurnal cycle in Figure 2 A-C for completeness but from there on restricted the analysis to daytime values. In Figure 2 this was clarified with grey shading the areas outside our daytime window and we further clarified this in the text.*

**Comment:** L415-6 - It's not clear what the authors mean by "extrapolated to". Do they simply mean assumed to be the same during the summer?
*Reply: Yes, the ratio was assumed to be the same in summer and winter.*
*Change: We clarified this in the revised manuscript.*

**Comment:** L433-434 - See previous comments regarding the superficial nature of the conclusion that the anomaly increased with increasing T and PAR. It would be far more useful if the authors could demonstrate that e.g. SMAP soil moisture content or VPD or … were not in fact the cause of the apparent correlation.
*Reply: See above discussion and clarifications on SMAP soil moisture data usage.*

**Comment:** L434 - The authors have not considered water availability at root depth. Please replace this term with something more appropriate.
*Reply: See above discussion and clarifications on SMAP soil moisture data usage - the term has been replaced with "precipitation and a coarse-scale satellite-based soil moisture product as a proxy for plant water availability"*

**Comment:** L447 - Is it big leaf or 5-layer?
*Reply: big leaf*

---

## Author Comment (AC2)

**Reviewer 2:**

SUMMARY
The authors present eddy-covariance VOC measurements over an urban footprint in Innsbruck, Austria. They examine the observed terpenoid (isoprene, monoterpenes, sesquiterpenes) fluxes in terms of tree species coverage within the flux footprint and in terms of the T and light dependencies driving emissions. They go on to compare results from two summers to assess interannual variability and the degree to which it can be understood mechanistically. There are not very many urban VOC flux datasets, and fewer still that explore interannual differences. I find the paper makes a useful contribution and is suitable for publication in ACP. Below I include some minor comments for the authors to consider.

GENERAL COMMENTS

**Comment:** 81-100, A nicely comprehensive summary of prior work, but quite long. Since the current paper doesn't address seasonal variability, it may be helpful to shorten this paragraph to only focus on those prior studies examining interannual changes.

*Response: We agree with the reviewer and removed the part about the seasonal variability. Change: Remove seasonal variability references to shorten the paragraph.*

**Comment:** 119, to avoid confusion please clarify if you mean into the NE/SW or out of the NE/SW

*Response: We thank the reviewer for this useful comment and changed the text accordingly.*
*Change: The text was changed to clarify that the wind is from NE during the day and from WS during the night.*

**Comment:** 121-122, please provide information here on the distribution of building heights within the flux footprint, and the degree to which the inlet is above versus within the roughness sublayer.
*Response: We have published this information before (Karl et al., BAMS, 2020, doi: 10.1175/BAMS-D-19-0270.1). Briefly, within 500 m from IAO, the mean building height is 17.3 m whereas the modal building height of about 19 m corresponds to the 5–7 story buildings, which are more important in terms of their form drag. For this reason, the displacement height, $z_d$, is estimated as 13.3 m (0.7 m × 19 m; e.g., Grimmond and Oke 1999). The roughness length, $z_0$, is 1.6 m.*

*Change: We have added the information about building height within the flux footprint.*

**Comment:** 137-146, it would be helpful to show some spectral analysis here to quantify how well the different frequency contributions to the fluxes were captured by the sampling system.
*Response: We thank reviewer 2 for this comment. We added a co-spectral analysis of the isopreniods to the Supplemental Information demonstrating very small attenuation loss of the sampling and measurement system.*

*Change: We inserted at the end of first paragraph of 2.2 Eddy covariance fluxes:*
*"Figure S1 shows the co-spectral response of the PTR-QiTOF-MS and inlet system. The loss of covariance of isoprenoids signals with vertical windspeed due to lowpass filtering is less than 4% (see Spectral analysis in Supplemental Information).*

**Comment:** 153, "Monoterpene and sesquiterpene eddy covariance fluxes are known to be purely temperature dependent". Not true. Some mono- and sesquiterpene emissions have been shown to also be light-dependent.
*Reply: We acknowledge that monoterpene and sesquiterpene emissions can also be light dependent. We agree that the word 'purely' in this context is not correct.*

*Change: We corrected the wording in section 2.2 on temperature dependent terpene emissions.*

**Comment:** 156-159, please indicate how these timescales were estimated

*Response: We thank the reviewer that there is more clarification needed and we added information on how the timescales were estimated in the text. The chemical lifetime was estimated according to measured ozone reaction rates with terpenes. The transport timescale was estimated by turbulence measurements (i.e. H/u\*).*

*Change: We added text describing how these timescales were estimated.*

**Comment:** 183-189, the validity of this ratio approach relies on the assumption that the ISO, MT and SQT emitters don't differ systematically in size (i.e. dry leaf weight). I imagine this is not strictly true. So some language here about this caveat is warranted.

*Response: We acknowledge this comment and added text clarifying that this is a caveat of our calculation as well as performed a sensitivity analysis doubling and halving the emission potentials of the highest 20 emitters to get a sense of the uncertainty due to unknown dry leaf weight differences in the trees and found that the average study area emission ratios changed on the order of 5-15%. This gives us a better estimate of the robustness of this analysis.*
*Change: We added the caveat as well as the robustness analysis in the text.*

**Comment:** 200, Very interesting! I would not have guessed that so much of the flux came from just 12 trees.
*Response: Yes, these trees have dominant emission potential and the differences in footprint density are overpowered by the emission strength of these emitters. The same number of trees don't account for 100% or the emissions but for the percentages as mentioned in the manuscript. The exact order of percent influence also varies between the two years due to footprint density differences but overall a large percentage of emissions arrive from the same trees.*
*Change: No changes*

**Comment:** 226-227, Guenther 2012 lists a light-dependent fraction of 0.5 for all sesquiterpenes. If you have evidence that sesquiterpene emissions are "mostly temperature dependent" it should be cited here. In the case of monoterpenes it is true that the most globally predominant emissions are mainly T-dependent but some individual species have >50% light dependence. So the extent to which "monoterpene emissions are mostly temperature-dependent" would depend on the monoterpene speciation in the flux footprint. Do you see any evidence for light-dependent MT/SQT emissions in your dataset?

*Response: It is true that Guenther et al. 2012 suggests a light - dependent fraction of 0.5 for sesquiterpenes. We changed the parameterization to include a possible light dependent fraction of sesquiterpenes. Since the PTR instrument can also not differentiate between*

*different monoterpene isomers, we can not fully exclude the possibility of light dependent terpene emissions. We estimate a ratio of 50% light dependence for monoterpenes as well using evidence from planted city trees and Guenther et al., 2012.*

*Change: We changed the parametrization for higher terpenes according to Guenther et al., 2012. The light dependent fraction for monoterpenes varies between 0.2 and 0.8, and for sesquiterpenes it is currently assumed to be 0.5. In addition to the simple temperature dependent formulation, we now use the temperature and light parameterization from Guenther et al., (2012) who prescribed a 50% light dependent fraction for SQT emissions. For Monoterpenes we estimate a light dependent fraction of 50%.*

**Comment:** 241-244, "Measured monoterpene and sesquiterpene measured fluxes at lower temperatures (280K-295K) were higher than the predicted values based on the Guenther et al. (1994) algorithm. This could be an indication that at lower temperatures other, non-biogenic sources contributed to monoterpene and sesquiterpene fluxes at this site." Along similar lines to the above, could this reflect partly a light-dependence?

*Response: Yes this is indeed a plausible explanation. We elaborate more on this issue. In particular we plot the residual of predicted vs observed MT fluxes and see a positive correlation with benzene fluxes, an anthropogenic tracer.*
*Change: We added a new figure and more discussion on the anthropogenic part of terpene emissions.*

**Comment:** 305-319, are any of the isoprene-emitters juvenile trees? I.e. could tree growth within the 3 years be relevant?

*Response: Rather unlikely, since just 8 % of the strong isoprene emitters were younger than 5 years in 2015.*
*Change: A corresponding sentence has been added to the revised manuscript.*

**Comment:** 365-366, "Mild to severe drought conditions would reduce isoprene emissions further and therefore could not explain an increased isoprene emission potential". This is confusing because paragraph (c) above discusses isoprene fluxes increasing under drought. Some more clarity in the arguments is needed.

*Response: It is true that drought parameterizations based on wilting points or similar would generally lower the isoprene emission potential. The reason why the isoprene emission potential can increase during the onset of drought is still debated and we have elaborated on this in our response to reviewer 1. Most likely higher emissions during the early phase of drought are attributed to changes in leaf temperature.*

*Change: We have clarified this paragraph accordingly.*

MINOR / TECHNICAL / WORDING SUGGESTIONS
27, suggest deleting "formation", it is redundant here
35, suggest "in predominantly isoprene-emitting forests"
50, "determined by PMF to mainly (60-70%) originate from vegetation"
51, "… isoprene, attributing it therefore"
58, "Whereas all the studies cited above…"
60, comma after "dilution"
60 & 82, period rather than colon

70, "as well as via storm water interception"
72, "are very plant-species dependent"
78-80, this sentence appears out of place
81, "even fewer such studies"
97, "is with 18% in July" is awkward
230, should say "Mean daytime maxima"

*Response: We thank the reviewer for these wording suggestions and changed the text.*
*Change: Changed text accordingly*

---

## Author Comment (AC3)

**Reviewer 3:**

In general, BVOC flux observations are a rare commodity. Even the limited observational datasets are mostly limited to forest environments. Kaser et al. present a city scale BVOC emissions from managed vegetation. They compared differences in BVOC emissions such as isoprene, monoterpenes, and sesquiterpenes from two years illustrating substantial differences in isoprene emissions but not monoterpene and sesquiterpene emissions. They have presented a thorough discussion for the potential causes. The discussion is particualy insightful to further explore the roles of managed vegetations in urban environments in local air quality. In summary, this manuscript is well written and would contribute to expand our knowledge in the atmospheric chemistry community. However, I would like to suggest a further detailed discussion on the differences in flux foot prints between 2015 and 2018 and their roles in differences in isoprene flux. In the 2015 footprint, a green space to the Southeast of the observational site (Figure 1) was exlcusively included and its potential role to the differences in isoprene emission could be highly insightful.

*Response: We thank reviewer 3 for this comment and clarified Figure 1. In the initial submission we showed only the 60% flux footprint for both years which could lead to the interpretation that the two years are quite different and the mentioned green space is only in one of the two years footprint. Figure 1 was now expanded to show all footprint density isolines from 30% to 90%. This clarifies that the two years footprints are not as different from each other and that the green space in the Southeast influences both years fluxes just to a somewhat different extent. We also extended the discussion on the footprint influence on the observed isoprene flux differences.*

*Change: Updated Figure 1 and text describing Figure 1 and footprint influence on isoprene fluxes.*